# Bridging the Sim-to-Real Gap from the Information Bottleneck Perspective

**Haoran He** [1,2]    **Peilin Wu** [1]    **Chenjia Bai** [3]    **Hang Lai** [1]    **Lingxiao Wang** [4]

**Ling Pan** [2]    **Xiaolin Hu** [5]    **Weinan Zhang** [1†]

[1] Shanghai Jiao Tong University    [2] Hong Kong University of Science and Technology
[3] Institute of Artificial Intelligence (TeleAI), China Telecom
[4] Northwestern University    [5] Tsinghua University

**Abstract:** Reinforcement Learning (RL) has recently achieved remarkable success in robotic control. However, most works in RL operate in simulated environments where privileged knowledge (e.g., dynamics, surroundings, terrains) is readily available. Conversely, in real-world scenarios, robot agents usually rely solely on local states (e.g., proprioceptive feedback of robot joints) to select actions, leading to a significant sim-to-real gap. Existing methods address this gap by either gradually reducing the reliance on privileged knowledge or performing a two-stage policy imitation. However, we argue that these methods are limited in their ability to fully leverage the available privileged knowledge, resulting in suboptimal performance. In this paper, we formulate the sim-to-real gap as an information bottleneck problem and therefore propose a novel privileged knowledge distillation method called the Historical Information Bottleneck (HIB). In particular, HIB learns a privileged knowledge representation from historical trajectories by capturing the underlying changeable dynamic information. Theoretical analysis shows that the learned privileged knowledge representation helps reduce the value discrepancy between the oracle and learned policies. Empirical experiments on both simulated and real-world tasks demonstrate that HIB yields improved generalizability compared to previous methods.

**Keywords:** Sim-to-Real, Information Bottleneck, Reinforcement Learning

## 1    Introduction

Reinforcement Learning (RL) has made significant advancements across various simulated environments (e.g., games [1], financial trading [2]), but its applications in real-world scenarios still remain a challenge. The primary obstacle is the *sim-to-real gap*, an inherent *mismatch* between simulated and real environments that cause policies learned in simulation to perform sub-optimally in the real world. Previous works tackle this problem through more realistic simulation [3, 4], adversarial training [5, 6], and domain randomization [7, 8] to minimize the mismatch. However, building high-quality simulators is difficult, and excessive introduced assumptions can lead to an overly conservative policy.

Recently, a more effective branch of methods proposes to utilize privileged knowledge to address the sim-to-real problem, where privileged knowledge is the information available in simulation (e.g., dynamics, surroundings, terrains) but inaccessible in real environments. Previous methods solve this problem via a two-stage policy distillation process [9]. Specifically, a teacher policy is first trained in the simulator with privileged states accessible, and then a student policy relying on local states (without privileged information) is trained by imitating the teacher policy. Nevertheless, such a two-stage paradigm is computationally expensive and requires careful design for imitation. An alternative approach involves gradually dropping the privileged information as the policy is trained [10] or conditioning the critic on privileged information [11]. However, these methods lack a theoretical understanding of the sim-to-real problem and do not fully exploit the available privileged information during training.

---

[†] Correspondence to: Weinan Zhang (wnzhang@sjtu.edu.cn).

8th Conference on Robot Learning (CoRL 2024), Munich, Germany.

In this paper, we present a representation-based approach, instead of policy distillation, to better utilize the privileged knowledge from simulation with a single-stage learning paradigm. Inspired by the Information Bottleneck (IB) method [12, 13], which learns a minimal sufficient representation $Z$ of a given input source $X$ with the target source $Y$, we propose a novel method called **H**istorical **I**nformation **B**ottleneck (HIB). Similar to the goal of the IB method, HIB aims to find a maximally compressed representation of the privileged knowledge while preserving sufficient information about the current environment for real-world decision-making. In particular, HIB takes advantage of historical information that contains previous local states and actions to learn a history representation, which is trained by maximizing the mutual information (MI) between the representation and the privileged knowledge. Theoretically, we show that maximizing such an MI term will minimize the privileged knowledge modeling error, reducing the discrepancy between the optimal value function and the learned value function. Furthermore, benefiting from the IB principle, we compress the decision-irrelevant information from the history and obtain a more robust representation. The IB objective is approximated by variational lower bounds to handle the high-dimensional state space.

In summary, our contributions are threefold: (i) We propose a novel policy generalization method called HIB that follows the IB principle to distill privileged knowledge from a fixed length of history. (ii) We provide a theoretical analysis of both the policy distillation methods and the proposed method, which shows that minimizing the privilege modeling error is crucial in learning a near-optimal policy. (iii) Empirically, we show that HIB learns robust representation in randomized RL environments and achieves better generalization performance in both simulated and real-world environments than state-of-the-art (SOTA) algorithms, including out-of-distribution test environments.

## 2 Related Work

**Sim-to-Real Transfer.** Transferring RL policies from simulation to reality is challenging due to the domain mismatch. To this end, the previous study hinges on domain randomization, which trains the policy under a wide range of environmental parameters and sensor noises [14, 15, 16, 17, 18]. However, domain randomization typically sacrifices the optimality for robustness, leading to an over-conservative policy [19]. To address this problem, various works perform privilege distillation by teacher-student learning [20, 21, 22, 23, 24], teacher demonstration exploration [25, 26, 27, 28] and teacher policy progressive imitation [29, 30]. However, these methods are sample inefficient due to the requirement of training an additional teacher policy. Aside from privileged policy, recent works exploit the privileged information by introducing privileged critic [31, 32, 11] or privileged world models [33, 34]. An alternative method gradually drops privileged knowledge [10]. Nevertheless, these methods are limited to fully leveraging historical knowledge for better generalization. Different from the above approaches, our method learns a privileged representation via informative historical trajectories from the IB perspective, resulting in better utilization of historical information.

**Information Bottleneck for RL.** The IB principle [35, 13] was initially proposed to trade off the accuracy and complexity of the representation in supervised learning. Specifically, IB maximizes the MI between representation and targets to extract useful features, while also compressing the irrelevant information by limiting the MI between representation and raw inputs [36, 37]. Recently, IB has been employed in RL to acquire a compact and robust representation. For example, Fan and Li [38] takes advantage of IB to learn task-relevant representation via a multi-view augmentation. Other methods [39, 40, 41] maximize the MI between representation and dynamics or value function, and restrict the information to encourage the encoder to extract only the task-relevant information. Unlike the previous works that neither tackle the policy generalization problem nor utilize historical information, HIB derives a novel objective based on IB, which aims to learn a robust representation of privileged knowledge from history while simultaneously removing redundant decision-irrelevant information.

## 3 Preliminaries

In this section, we briefly introduce the problem definition and the corresponding notations used throughout this paper. We give the definition of privileged knowledge in robot learning as follows.

**Definition 1** (Privileged Knowledge). *Privileged knowledge is the hidden state that is inaccessible in the real environment but can be obtained in the simulator, e.g., surrounding heights, terrain types, morphology parameters like length of legs, and dynamic parameters like friction and damping. An oracle (teacher) policy is defined as the optimal policy with privileged knowledge visible.*

In this paper, we extend the concept of the sim-to-real gap to a general policy generalization problem with a knowledge gap. We define the MDP as $\mathcal{M} = (\mathcal{S}^l, \mathcal{S}^p, \mathcal{A}, P, r, \gamma)$, where $[s^l, s^p] = s^o$ represents the oracle state $s^o$ that contains $s^l \in \mathcal{S}^l$ (i.e., the local state space) and $s^p \in \mathcal{S}^p$ (i.e., the privileged state space), where $s^p$ contains privileged knowledge defined in Definition 1. $\mathcal{A}$ is the action space. The transition function $P(s^o_{t+1}|s^o_t, a_t)$ and reward function $r(s^o, a)$ follows the ground-truth dynamics based on the oracle states. Based on the MDP, we define two policies: $\pi(a|s^l, s^p)$ and $\hat{\pi}(a|s^l)$, for the simulation and real world, respectively. Specifically, $\pi(a|s^l, s^p)$ is a privileged policy that can access the privileged knowledge, which is only accessible in the simulator. In contrast, $\hat{\pi}(a|s^l)$ is a local policy without accessing the privileged knowledge throughout the interaction process, which is common in the real world. A thorough discussion about our problem and Partially Observable MDP (POMDP) [42, 43] is provided in Appendix A.1.

Based on the above definition, our objective is to find the optimal local policy $\hat{\pi}^*$ based on the local state $s^l$ that maximizes the expected return, denoted as

$$\hat{\pi}^* := \arg\max_{\hat{\pi}} \mathbb{E}_{a_t \sim \hat{\pi}(\cdot|s^l_t)} \Big[ \sum_{t=0}^{\infty} \gamma^t r(s^o_t, a_t) \Big]. \tag{1}$$

## 4 Theoretical Analysis & Motivation

### 4.1 Value Discrepancy for Policy Generalization

In this section, we give a theoretical analysis of traditional oracle policy imitation algorithms [44]. Note that previous works give similar analyses but in different formulations [22, 27, 30]. Specifically, the local policy is learned by imitating the optimal oracle policy $\pi^*$. We denote the optimal value function of the policy $\pi^*$ learned with oracle states $s^o = [s^l, s^p]$ as $Q^*(s^l, s^p, a)$, and the value function of policy learned with local states as $\hat{Q}^{\hat{\pi}}(s^l, a)$. The following theorem analyzes the relationship between the value discrepancy and the policy imitation error in a finite MDP setting.

**Theorem 1** (Policy imitation discrepancy). *The value discrepancy between the optimal value function with privileged knowledge and the value function with the local state is bounded as*

$$\sup_{s^l, s^p, a} \big|Q^*(s^l, s^p, a) - \hat{Q}^{\hat{\pi}}(s^l, a)\big| \leq \frac{2\gamma r_{\max}}{(1-\gamma)^2} \epsilon_{\hat{\pi}}, \tag{2}$$

*where*

$$\epsilon_{\hat{\pi}} = \sup_{s^l, s^p} D_{\text{TV}}\big(\pi^*(\cdot|s^l, s^p)\|\hat{\pi}(\cdot|s^l)\big) \tag{3}$$

*is the policy divergence between $\pi^*$ and $\hat{\pi}$, and $r_{\max}$ is the maximum reward in each step.*

The proof is given in Appendix A.2. Theorem 1 shows that minimizing the total variation (TV) distance between $\pi^*(\cdot|s^l, s^p)$ and $\hat{\pi}(\cdot|s^l)$ reduces the value discrepancy. However, minimizing $\epsilon_{\hat{\pi}}$ can be more difficult than ordinary imitation learning where $\pi^*$ and $\hat{\pi}$ have the same state space. Specifically, if $\pi^*$ and $\hat{\pi}$ have the same inputs, $D_{\text{TV}}\big(\pi^*(\cdot|s)\|\hat{\pi}(\cdot|s)\big)$ will approach zero with sufficient model capacity and large iteration steps, at least in theory. In contrast, due to the lack of privileged knowledge in our problem, the policy error term in Eq. (3) can still be large after optimization as $\pi^*(\cdot|s^l, s^p)$ and $\hat{\pi}(\cdot|s^l)$ have different inputs.

Previous works [21, 20] try to use historical trajectory

$$h_t = \{s^l_t, a_{t-1}, s^l_{t-1}, \ldots, a_{t-k}, s^l_{t-k}\} \tag{4}$$

with a fixed length to help infer the oracle policy, and the policy imitation error becomes $D_{\text{TV}}\big(\pi^*(\cdot|s^l, s^p)\|\hat{\pi}(\cdot|s^l, h)\big)$. However, without appropriately using the historical information, imitating a well-trained oracle policy can still be difficult for an agent with limited capacity, which results in suboptimal performance. Meanwhile, such two-stage policy imitation methods are also sample-inefficient.

### 4.2 Privilege Modeling Discrepancy

To address the above challenges, we raise an alternative theoretical motivation to relax the requirement of the oracle policy. Specifically, we quantify the discrepancy between the optimal value function and the learned value function based on the error bound in reconstructing the privileged state $s_t^p$ via historical information $h_t$, which eliminates the reliance on the oracle policy and makes our method a *single-stage* distillation algorithm. Specifically, we define a density model $\hat{P}(s_t^p|h_t)$ to predict the privileged state based on history $h_t$ in Eq. (4). Then the predicted privileged state can be sampled as $\hat{s}_t^p \sim \hat{P}(\cdot|h_t)$. In policy learning, we concatenate the local state $s^l$ and the predicted $\hat{s}^p$ as input. The following theorem gives the value discrepancy of $Q^*$ and the value function $\hat{Q}$ with the predicted $\hat{s}^p$.

**Theorem 2** (Privilege modeling discrepancy). *Let the divergence between the distribution of privileged state model $\hat{P}(s_{t+1}^p|h_{t+1})$ and the true distribution of privileged state $P(s_{t+1}^p|h_{t+1})$ be bounded as*

$$\epsilon_{\hat{P}} = \sup_{t \geq t_0} \sup_{h_{t+1}} D_{\text{TV}}\big(P(\cdot \mid h_{t+1}) \,\|\, \hat{P}(\cdot \mid h_{t+1})\big). \tag{5}$$

*Then the performance discrepancy bound between the optimal value function with $P$ and the value function with $\hat{P}$ holds, as*

$$\sup_{t \geq t_0} \sup_{s^l, s^p, a} |Q^*(s_t^l, s_t^p, a_t) - \hat{Q}_t(s_t^l, \hat{s}_t^p, a_t)| \leq \frac{\Delta_{\mathbb{E}}}{(1-\gamma)} + \frac{2\gamma r_{\max}}{(1-\gamma)^2}\epsilon_{\hat{P}}, \tag{6}$$

*where $\Delta_{\mathbb{E}} = \sup_{t \geq t_0} \big\|Q^* - \mathbb{E}_{s_t^p \sim P(\cdot|h_t)}[Q^*]\big\|_{\infty} + \big\|\hat{Q} - \mathbb{E}_{\hat{s}_t^p \sim \hat{P}(\cdot|h_t)}[\hat{Q}]\big\|_{\infty}$ is the difference in the same value function with sampled $s_t^p$ and the expectation of $s_t^p$ conditioned on $h_t$.*

We defer the detailed proof in Appendix A.3. In $\Delta_{\mathbb{E}}$, we consider using a sufficiently long (i.e., by using history $t$ greater than some $t_0$) and informative (i.e., by extracting useful features) history to make $\Delta_{\mathbb{E}}$ small in practice. We remark that $\Delta_{\mathbb{E}}$ captures the inherent difficulty of learning without privileged information. The error is small if privileged information is near deterministic given the history, or if the privileged information is not useful given the history. $\epsilon_{\hat{P}}$ measures the model discrepancy in the worst case with an informative history. In the next section, we provide an instantiation method inspired by Theorem 2.

## 5 Methodology

In this section, we propose a practical algorithm named HIB to perform privilege distillation via a historical representation. HIB only acquires the oracle state without an oracle policy in training. In evaluation, HIB relies on the local state and the learned historical representation to choose actions.

### 5.1 Reducing the Discrepancy via MI

Theorem 2 indicates that minimizing $\epsilon_{\hat{P}}$ yields a tighter performance discrepancy bound. We then start by analyzing the privilege modeling discrepancy $\epsilon_{\hat{P}}$ in Eq. (5). We denote the parameter of $\hat{P}_\phi$ by $\phi$, then the optimal solution $\phi^*$ can be obtained by minimizing the TV divergence for $\forall t$, as

$$\phi^* = \arg\min_\phi D_{\text{TV}}\big(P(\cdot|h_t)\|\hat{P}_\phi(\cdot|h_t)\big) = \arg\min_\phi D_{\text{KL}}\big(P(\cdot|h_t)\|\hat{P}_\phi(\cdot|h_t)\big) \tag{7}$$

$$= \arg\max_\phi \mathbb{E}_{p(s_t^p, h_t)}\big[\log \hat{P}_\phi(s_t^p|h_t)\big] \triangleq \arg\max_\phi I_{\text{pred}}, \tag{8}$$

where the true distribution $P(\cdot|h_t)$ is irrelevant to $\phi$, and we convert the TV distance to the KL distance in Eq. (7) by following Pinsker's inequality. Since $h_t$ is usually high-dimensional, which is of linear complexity with respect to time, it is necessary to project $h_t$ in a representation space and then predict $s_t^p$. Thus, we split the parameter of $\hat{P}_\phi$ as $\phi = [\phi_1, \phi_2]$, where $\phi_1$ aims to learn a historical representation $z = f_{\phi_1}(h_t)$ first, and $\phi_2$ aims to predict the distribution $\hat{P}_{\phi_2}(z)$ of privileged state (e.g., a Gaussian). In the following, we show that maximizing $I_{\text{pred}}$ is closely related to maximizing the MI between the historical representation and the privileged state. In particular, we have

$$\begin{aligned}
I_{\text{pred}} &= \mathbb{E}_{p(s_t^p, h_t)}\big[\log \hat{P}_{\phi_2}\big(s_t^p|f_{\phi_1}(h_t)\big)\big] \\
&= \mathbb{E}_{p(s_t^p, h_t)}\big[\log P\big(s_t^p|f_{\phi_1}(h_t)\big)\big] - D_{\text{KL}}[P\|\hat{P}] = -\mathcal{H}\big(S_t^p|f_{\phi_1}(H_t)\big) - D_{\text{KL}}[P\|\hat{P}] \quad (9) \\
&= I\big(S_t^p; f_{\phi_1}(H_t)\big) - \mathcal{H}(S_t^p) - D_{\text{KL}}[P\|\hat{P}] \leq I\big(S_t^p; f_{\phi_1}(H_t)\big),
\end{aligned}$$

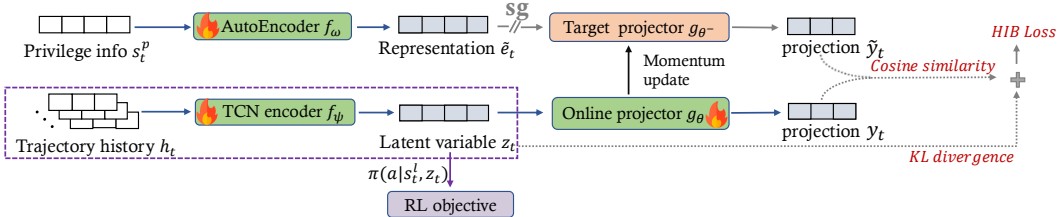

Figure 1: Overall training framework. HIB adopts the IB principle to recover privileged knowledge from a fixed length of local history information. The RL objective also provides gradients to the history encoder $f_\psi$, implying that the learned representation can be combined with any RL algorithm effectively.

where we denote the random variables for $s_t^p$ and $h_t$ by $S_t^p$ and $H_t$, respectively. In Eq. (9), the upper bound is obtained by the non-negativity of the Shannon entropy and KL divergence. The bound is tight since the entropy of the privileged state $\mathcal{H}(S_t^p)$ is usually constant, and $D_{\mathrm{KL}}(P\|\hat{P})$ can be small when we use a variational $\hat{P}_\phi$ with an expressive network.

According to Eq. (9), maximizing the predictive objective $I_{\mathrm{pred}}$ is closely related to maximizing the MI between $S_t^p$ and $f_{\phi_1}(H_t)$. In HIB, to address the difficulty of reconstructing the raw privileged state that can be noisy and high-dimensional in $I_{\mathrm{pred}}$ objective, we adopt contrastive learning [45] as an alternative variational approximator [46] to approximate MI in representation space. Moreover, HIB restricts the capacity of representation to remove decision-irrelevant information from the history, which resembles the IB principle [35] in information theory.

## 5.2 Historical Information Bottleneck

To optimize the MI in Eq. (9) via a contrastive objective [47], we introduce a historical representation $z_t \sim f_\psi(h_t)$ to extract useful features that contain privileged information from a long historical vector $h_t$, where $f_\psi$ is a temporal convolution network (TCN) [48] that captures long-term information along the time dimension. We use another notation $\psi$ to distinguish it from the predictive encoder $\phi_1$ in Eq. (9), since the contrastive objective and predictive objective $I_{\mathrm{pred}}$ learn distinct representations by optimizing different variational bounds. In our IB objective, the input variable is $H_t$, and the corresponding target variable is $S_t^p$. Our objective is to maximize the MI term $I(Z_t; \mathcal{S}_t^p)$ while minimizing the MI term $I(H_t; Z_t)$ with $Z_t = f_\psi(H_t)$, which takes the form of

$$\min -I(Z_t; S_t^p) + \alpha I(H_t; Z_t), \tag{10}$$

where $\alpha$ is a Lagrange multiplier. The $I(Z_t; S_t^p)$ term quantifies the amount of information about the privileged knowledge preserved in $Z_t$, and the $I(H_t; Z_t)$ term is a regularizer that controls the complexity of representation learning. With a well-tuned $\alpha$, we do not discard useful information that is relevant to the privileged knowledge.

We minimize the MI term $I(H_t; Z_t)$ in Eq. (10) by minimizing the following tractable upper bound:

$$I(H_t; Z_t) = \mathbb{E}_{p(z_t, h_t)}\left[\log \frac{p(z_t|h_t)}{p(z_t)}\right] = \mathbb{E}_{p(z_t, h_t)}\left[\log \frac{p(z_t|h_t)}{q(z_t)}\right] - D_{\mathrm{KL}}[p(z_t)\|q(z_t)]$$
$$\leq D_{\mathrm{KL}}[p(z_t|h_t)\|q(z_t)], \tag{11}$$

where the inequality follows the non-negativity of the KL divergence, and $q(z_t)$ is an approximation of the marginal distribution of $Z_t$. We follow Alemi et al. [36] and use a spherical Gaussian $q(z_t) = \mathcal{N}(0, I)$ as an approximation.

One can maximize the MI term $I(Z_t; S_t^p)$ in Eq. (10) based on the contrastive objective [47]. Specifically, for a given $s_t^p$, the positive sample $z_t \sim f_\psi(h_t)$ is the feature of corresponding history in timestep $t$, and the negative sample $z^-$ can be extracted from randomly sampled historical vectors. However, considering unresolved trade-offs involved in negative sampling [49, 50], we simplify the contrastive objective by removing negative sampling. In HIB, the contrastive loss becomes a cosine similarity with only positive pairs, and we empirically find that the performance does not decrease. Such a simplification was also adopted by recent contrastive methods for RL [51, 52].

We adopt a two-stream architecture to learn $z_t$, including an *online* and a *target* network. Each network contains an encoder and a projector, as shown in Fig. 1. The online network is trained to use history to predict the corresponding privilege representation. Given a pair of a history sequence and privileged state $(h_t, s_t^p)$, we obtain $\tilde{e}_t = f_\omega(s_t^p)$ with an encoder $f_\omega$ to get the representation of $s_t^p$. Here, $f_\omega$ is used to project $s_t^p$ into the same dimensional space as $z_t$, so $f_\omega$ can be a simple AutoEncoder [53] trained by a reconstruction loss $\mathcal{L}_{\text{rec}} = \|s_t^p - D_{\omega'}(f_\omega(s_t^p))\|^2$, where $D_{\omega'}$ is an additional trained decoder (see Appendix C.1 for details of implementation). Then we use TCN as the history encoder $f_\psi$ to learn the latent representation $z_t \sim f_\psi(\cdot|h_t)$. As $\tilde{e}_t$ and $z_t$ have the same dimensions, the projectors share the same architecture. The online projector $g_\theta$ outputs $y_t = g_\theta(z_t)$ and the target projector $g_{\theta^-}$ outputs $\tilde{y}_t = g_{\theta^-}(\tilde{e}_t)$. We use the following cosine similarity loss between $y_t$ and $\tilde{y}_t$, and use stop gradient (sg[$\cdot$]) for the target value $\tilde{y}$, as

$$\mathcal{L}_{\text{sim}} = -\sum_{y_t, \hat{y}_t} \left(\frac{y_t}{\|y_t\|_2}\right)^\top \left(\frac{\text{sg}[\tilde{y}_t]}{\|\text{sg}[\tilde{y}_t]\|_2}\right). \tag{12}$$

To prevent collapsed solutions in the two-stream architecture, we follow previous architectures [54] by using a momentum update for the target network to avoid collapsed solutions. Specifically, the parameter of the target network $\theta^-$ takes an exponential moving average of the online parameters $\theta$ with a factor $\tau \in [0, 1]$, as $\theta^- \leftarrow \tau\theta^- + (1-\tau)\theta$.

At each training step, we perform a stochastic optimization step to minimize $\mathcal{L}_{\text{sim}}$ with respect to $\theta$ and $\psi$. Meanwhile, we learn an RL policy $\pi(a|s_t^l, z_t)$ based on the historical representation $z_t$, and the RL objective is also used to train the TCN encoder $f_\psi$. The dynamics are summarized as

$$\theta \leftarrow \text{optimizer}(\theta, \nabla_\theta \mathcal{L}_{\text{sim}}), \psi \leftarrow \text{optimizer}(\psi, \nabla_\psi(\lambda_1 \mathcal{L}_{\text{sim}} + \lambda_2 \mathcal{L}_{\text{KL}} + \mathcal{L}_{\text{RL}}(s_t^l, z_t)), \omega \leftarrow \text{optimizer}(\omega, \nabla_\omega \mathcal{L}_{\text{rec}}) \tag{13}$$

where $\mathcal{L}_{\text{KL}} = D_{\text{KL}}(f_\psi(h_t)\|\mathcal{N}(0, I))$ is the IB term in Eq. (11) that controls the latent complexity, and $\mathcal{L}_{\text{RL}}$ is the loss function for an arbitrary RL algorithm. We use the Adam optimizer [55]. We summarize our algorithm in Alg. 1 in Appendix C.1.

## 6 Experiments

### 6.1 Benchmarks and Compared Methods

To quantify the generalizability of the proposed HIB, we conduct experiments in simulated environments that include multiple domains for a comprehensive evaluation, and also the legged robot locomotion task to evaluate the generalizability in sim-to-real transfer.

**Privileged DMC Benchmark.** We conduct experiments on DeepMind Control Suite (DMC) [56] with manually defined privileged information, which contains *dynamic parameters* such as friction and torque strength, and *morphology parameters* such as the lengths of specific legs. The privileged knowledge is *only* visible in the training process. Following Benjamins et al. [57], we randomize the privilege parameters at the beginning of each episode, and the randomization range can be different for training and testing. Specifically, we choose three randomization ranges for varied difficulty levels, i.e., *ordinary, o.o.d.*, and *far o.o.d.*. The *ordinary* setting means that the test environment has the same randomization range as in training, while *o.o.d.* and *far o.o.d.* indicate that the randomization ranges are larger with different degrees, causing test environments being out-of-distribution compared with training environment. The detailed setup can be found in Appendix B.1. For *dynamic parameters*, we evaluate the algorithms in three different domains, namely *pendulum*, *finger spin*, and *quadruped walk*; For *morphology parameters*, we evaluate the algorithms in *quadruped walk* with varying lengths of front legs and back legs. Both settings cover various difficulties ranging from *ordinary* to *far o.o.d.*. This benchmark is denoted as the DMC benchmark in the following for simplicity.

**Sim-to-Real in Blind Legged Robot.** This experiment is conducted on a quadrupedal robot. In this domain, privileged knowledge is defined as terrain information (e.g. heights of surroundings) of the environment and dynamic information such as friction, mass, and damping of the quadrupedal robot. We leverage the Isaac Gym simulator [58] for training, which also provides multi-terrain simulation (e.g., slopes, stairs, and discrete obstacles). Details can be found in Appendix B.2. For experiments in the real robot, we utilize the Unitree A1 robot [59] to facilitate real-world deployment.

Table 1: Evaluated episodic return achieved by HIB and baselines on DMC tasks. SAC is adopted as the basic RL algorithm. We report the mean and standard deviation for 100K steps. Variances are large because dynamic parameters are changed for each episode. We refer to Appendix B.1 for the details.

| Domain | Testing Difficulty | Teacher | HIB (ours) | DR | Student (RMA) | DreamWaQ | Dropper |
|---|---|---|---|---|---|---|---|
| Pendulum | ordinary | $-98.29_{\pm80.41}$ | $-110.33_{\pm89.67}$ | $-206.33_{\pm259.66}$ | $\mathbf{-107.69_{\pm90.87}}$ | $-118.09_{\pm115.46}$ | $-204.90_{\pm223.74}$ |
| | o.o.d. | $-251.16_{\pm384.52}$ | $\mathbf{-271.25_{\pm355.96}}$ | $-502.58_{\pm543.67}$ | $-401.39_{\pm504.42}$ | $-315.39_{\pm431.10}$ | $-436.82_{\pm473.84}$ |
| | far o.o.d. | $-660.98_{\pm535.22}$ | $\mathbf{-671.83_{\pm531.42}}$ | $-800.62_{\pm583.95}$ | $-729.69_{\pm551.88}$ | $-677.20_{\pm509.85}$ | $-674.43_{\pm520.75}$ |
| Finger Spin | ordinary | $826.19_{\pm152.61}$ | $\mathbf{714.06_{\pm233.85}}$ | $529.32_{\pm414.99}$ | $657.00_{\pm411.07}$ | $641.08_{\pm348.12}$ | $569.56_{\pm308.62}$ |
| | o.o.d. | $793.18_{\pm196.02}$ | $\mathbf{669.21_{\pm254.22}}$ | $460.79_{\pm417.85}$ | $649.58_{\pm405.97}$ | $618.72_{\pm318.64}$ | $551.61_{\pm318.03}$ |
| | far o.o.d. | $663.73_{\pm292.41}$ | $\mathbf{645.42_{\pm248.03}}$ | $453.00_{\pm396.03}$ | $598.76_{\pm409.50}$ | $605.01_{\pm356.14}$ | $518.05_{\pm293.88}$ |
| Quadruped Walk | ordinary | $973.34_{\pm30.12}$ | $\mathbf{946.72_{\pm31.43}}$ | $224.02_{\pm27.57}$ | $863.83_{\pm36.74}$ | $877.04_{\pm40.16}$ | $334.45_{\pm380.94}$ |
| | o.o.d. | $945.60_{\pm42.28}$ | $\mathbf{927.91_{\pm53.05}}$ | $203.21_{\pm38.98}$ | $829.27_{\pm60.37}$ | $820.45_{\pm58.90}$ | $287.68_{\pm352.45}$ |
| | far o.o.d. | $922.49_{\pm50.81}$ | $\mathbf{904.72_{\pm85.76}}$ | $164.79_{\pm64.32}$ | $793.41_{\pm89.67}$ | $801.68_{\pm95.83}$ | $286.00_{\pm347.52}$ |

Table 2: Evaluated episodic return achieved by HIB and baselines on *quadruped walk* task with morphology shifts. The lengths of the front legs and back legs are changed episodically. See Appendix B.1 for more details.

| Changed Legs | Testing Difficulty | Teacher | HIB (ours) | DR | Student (RMA) | DreamWaQ | Dropper |
|---|---|---|---|---|---|---|---|
| Front | ordinary | $932.99_{\pm38.67}$ | $\mathbf{912.92_{\pm55.19}}$ | $305.21_{\pm39.55}$ | $860.78_{\pm130.18}$ | $732.60_{\pm104.81}$ | $244.44_{\pm366.20}$ |
| | o.o.d. | $914.26_{\pm54.36}$ | $\mathbf{896.69_{\pm62.72}}$ | $300.94_{\pm27.32}$ | $851.79_{\pm144.73}$ | $728.37_{\pm102.04}$ | $238.84_{\pm357.67}$ |
| | far o.o.d. | $861.92_{\pm99.11}$ | $\mathbf{867.18_{\pm96.68}}$ | $294.50_{\pm32.07}$ | $814.52_{\pm178.59}$ | $721.16_{\pm126.57}$ | $220.11_{\pm340.13}$ |
| Back | ordinary | $926.73_{\pm42.32}$ | $\mathbf{914.05_{\pm52.02}}$ | $66.93_{\pm97.41}$ | $860.11_{\pm135.18}$ | $733.21_{\pm121.31}$ | $230.99_{\pm380.79}$ |
| | o.o.d. | $906.63_{\pm59.12}$ | $\mathbf{903.93_{\pm51.39}}$ | $65.30_{\pm90.56}$ | $855.98_{\pm126.60}$ | $728.49_{\pm101.54}$ | $180.95_{\pm337.38}$ |
| | far o.o.d. | $834.72_{\pm118.37}$ | $812.08_{\pm81.55}$ | $61.08_{\pm88.72}$ | $\mathbf{816.99_{\pm173.98}}$ | $727.84_{\pm115.93}$ | $150.39_{\pm296.54}$ |

**Baselines.** We compare HIB to the following baselines. (i) **Teacher** policy is learned by oracle states with privileged knowledge. (ii) **Student** policy follows RMA [21] that mimics the teacher policy through supervised learning, with the same architecture as the history encoder in HIB. We remark that student needs a two-stage training process to obtain the policy. (iv) **DreamWaQ** follows the implementation in Nahrendra et al. [31], conditioning the critic on privileged information and utilizing history trajectories to learn a latent variable $z$ for action. (iv) **Dropper** is implemented according to Li et al. [10], which gradually drops the privileged information and finally converts to a normal agent that only takes local states as input. (v) **DR** agent utilizes domain randomization for generalization and is directly trained with standard RL algorithms with local states as input.

## 6.2    Simulation Comparison

For varying dynamic parameters, the results are shown in Table 1. Our method achieves the best performance in almost all test environments, closely matching the returns of the oracle (teacher) method. The results in Table 2 demonstrate that HIB can also generalize to unseen environments with morphology shifts, highlighting its superior capability which benefits from our proposed IB-style training. While the student baseline can perform slightly better than HIB, it exhibits larger variance because its performance heavily relies on the teacher policy and its exploration ability is restricted. The results on the Legged Robot benchmark (Table 3) further verify the advantage of HIB. Our method outperforms the two strongest baselines, student and DreamWaQ, on the three most challenging terrains: stairs, discrete obstacles, and slopes. From the simulation results, HIB can be seamlessly combined with different RL algorithms and generalizes well across different domains and tasks, demonstrating the efficacy and high scalability of HIB.

Table 3: Success rates and average distances achieved by HIB and baselines of legged robot for 1000 steps evaluated in the Isaac-Gym simulator. Results are averaged over 1000 trajectories with different difficulties. Reward designs follow [3], and environmental details can be achieved in Appendix B.2.

| | Success Rate (%) ↑ | | | Average Distance (m) ↑ | | |
|---|---|---|---|---|---|---|
| | Stairs | Discrete Obstacles | Slope | Stairs | Discrete Obstacles | Slope |
| Student (RMA) | 94.2 | 82.1 | 96.1 | $7.96_{\pm1.21}$ | $7.67_{\pm2.14}$ | $8.69_{\pm1.09}$ |
| DreamWaQ | 94.1 | 81.3 | 97.1 | $8.09_{\pm1.34}$ | $7.51_{\pm2.38}$ | $8.78_{\pm0.95}$ |
| HIB (ours) | **96.7** | **85.8** | **97.7** | $\mathbf{8.67_{\pm1.16}}$ | $\mathbf{7.84_{\pm2.07}}$ | $\mathbf{9.04_{\pm1.12}}$ |
| Teacher | 98.0 | 86.9 | 98.3 | $8.88_{\pm1.09}$ | $8.21_{\pm2.01}$ | $9.24_{\pm1.06}$ |

## 6.3    Visualization and Ablation Study

To investigate the ability of HIB in modeling the privileged knowledge, we visualize the latent representation learned by HIB and *student* via dimensional reduction with t-SNE [60]. We also visualize the true privileged information for comparison. The visualization is conducted in *finger spin* task, and the results are given in Fig. 3. We find that the *student* agent can only recover part of the privileged knowledge that covers the bottom left and upper right of the true privilege distribution. In

contrast, the learned representation of HIB has almost the same distribution as privileged information. This may help explain why our method outperforms other baselines and generalizes to o.o.d. scenarios without significant performance degradation.

We conduct an ablation study for each component of HIB to verify their effectiveness. Specifically, we design the following variants to compare with. (i) **HIB-w/o-ib** only uses RL loss to update history encoder $f_\psi$, which is similar to a standard recurrent neural network policy. (ii) **HIB-w/o-rl** only uses HIB loss to update the history encoder without the RL objective. (iii) **HIB-w/o-proj** drops the projectors in HIB and directly compute cosine similarity loss between $z_t$ and $\tilde{e}_t$. (iv) **HIB-contra** employs contrastive loss [61] instead of cosine similarity, using a score function that assigns high scores to positive pairs and low scores to negative pairs.

From the result in Fig. 4, we observe that HIB-w/o-ib almost fails and HIB-w/o-rl can get relatively high scores, which signifies that both HIB loss and RL loss are important for the agent to learn a well-generalized policy, especially the HIB loss. The HIB loss helps the agent learn a historical representation that contains privileged knowledge for better generalization. Furthermore, projectors and the momentum update mechanism are also crucial in learning robust and effective representation. Moreover, HIB-contra performs well at the beginning but fails later, which indicates that the contrastive objective requests constructing valid negative pairs and learning a good score function, which is challenging in the general state-based RL setting.

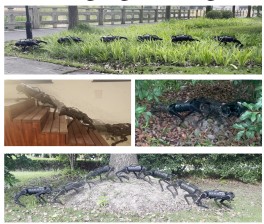

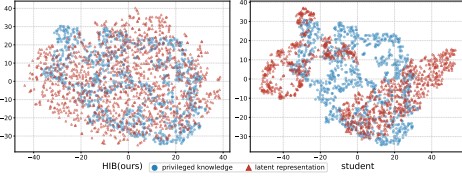

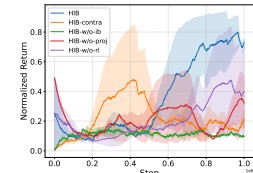

Figure 2: A Unitree A1 traversing different terrains.

Figure 3: t-SNE visualization for the privilege representation and the learned latent representation of history encoder in *finger spin*.

Figure 4: Comparison of different HIB variants in *quadruped walk*.

## 6.4 Real-world Application

To further evaluate the generalizability of HIB in the real world, we deploy the HIB policy trained in the Legged Robot benchmark on a real-world A1 robot without any fine-tuning. Detailed setup is referred to Appendix B.3. Note that the policy is directly run on the A1 hardware, and the local state is read or estimated from the onboard sensors and IMU, making the real-world control noisy and challenging. Fig. 2 shows the snapshots of our HIB

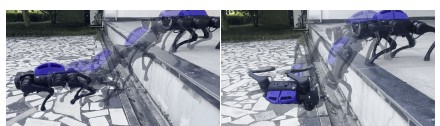

(a) w/ HIB      (b) w/o HIB (DreamWaQ)

Figure 5: HIB successfully handles high stairs. See more examples in our supplementary video.

agent traversing different terrains. The agent can generalize to different challenging terrains with stable control behavior and there is **no failure** in the whole experimental process. Fig. 5 further showcases the advantages of our proposed HIB, enabling the agent to navigate down high stairs with a $0.6$m height, achieving a **100**% success rate in 10 trials, while the strongest baseline (e.g. DreamWaQ) completely fail. These real experiments demonstrate that HIB enhances the ability to bridge the sim-to-real gap without any additional tuning in real environments.

## 7 Conclusion and Limitation

We propose a novel privileged knowledge distillation method based on the Information Bottleneck to narrow the knowledge gap between local and oracle RL environments. In particular, the proposed two-stream model design and HIB loss help reduce the performance discrepancy given in our theoretical analysis. Our experimental results on both simulated and real-world environments show that (i) HIB learns robust representations to reconstruct privileged knowledge from local historical trajectories and boosts the RL agent's performance, and (ii) HIB can achieve improved generalizability in out-of-distribution environments compared to previous methods. However, HIB is limited in recovering multi-modal privileged knowledge (e.g., RGB images), which is more high-dimensional and complex.

## Acknowledgments

This work is partially supported by Shanghai Municipal Science and Technology Major Project (2021SHZDZX0102) and National Natural Science Foundation of China (62322603 & 62076161). We also thank the anonymous reviewers for their valuable suggestions.

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

# A  Theorems and Proofs

## A.1  POMDP and Sim-to-Real Problems

We summarize common points and differences between POMDP and our setting as follows.

1. The transition function and reward function for both problems follows the ground-truth dynamics of the environment.

2. In POMDP, the agent cannot access the oracle state in both training and evaluation, while in sim-to-real adaptation, the agent can access the oracle in training (in the simulation).

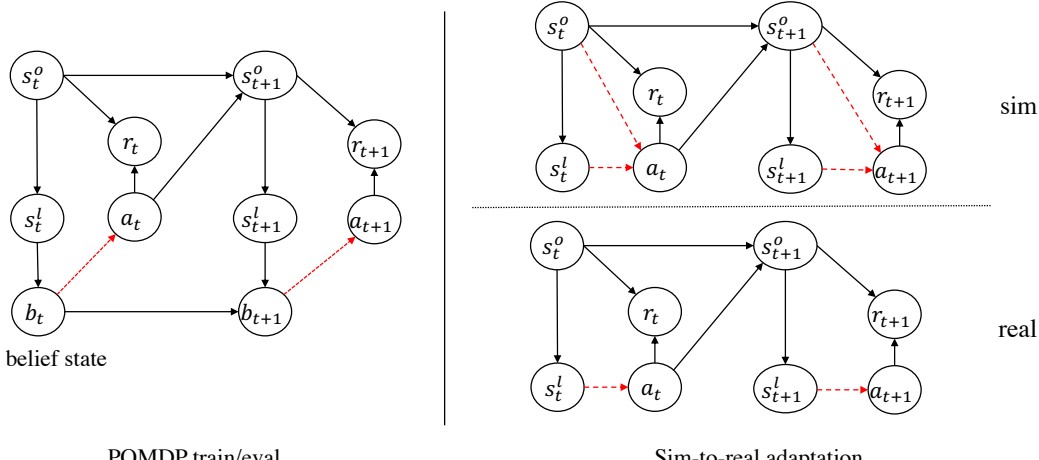

Figure 6: The difference between POMDP setting and the sim-to-real problem.

## A.2  Proof of Theorem 1

We begin by deriving the imitation discrepancy bound in an ordinary MDP, as shown in the following theorem.

**Theorem A.1** (General oracle imitation discrepancy bound). *Let the policy divergence between the oracle policy $\pi^*$ and the learned policy $\hat{\pi}$ is $\epsilon_\pi = \sup_s D_{\mathrm{TV}}(\pi^*(\cdot|s)\|\hat{\pi}(\cdot|s))$. Then the difference between the optimal action value function $Q^*$ of $\pi^*$ and the action value function $\hat{Q}$ of $\hat{\pi}$ is bounded as*

$$\sup_{s,a} \left| Q^*(s,a) - \hat{Q}(s,a) \right| \leq \frac{2\gamma r_{\max}}{(1-\gamma)^2} \epsilon_\pi, \tag{14}$$

*where $\gamma$ is the discount factor, and $r_{\max}$ is the maximum reward in each step.*

*Proof.* For any $(s,a)$ pair that $s \in \mathcal{S}, a \in \mathcal{A}$, the difference between $Q^*(s,a)$ and $\hat{Q}(s,a)$ is

$$\begin{aligned}
&Q^*(s,a) - \hat{Q}(s,a) \\
&= r(s,a) + \gamma \mathbb{E}_{s' \sim P(s,a), a' \sim \pi^*(\cdot|s')}\left[Q^*(s',a')\right] - r(s,a) - \gamma \mathbb{E}_{s' \sim P(s,a), a'' \sim \hat{\pi}(\cdot|s')}\left[\hat{Q}(s',a'')\right] \\
&= \gamma \sum_{s',a'} P(s'|s,a)\left[\pi^*(a'|s')Q^*(s',a') - \hat{\pi}(a'|s')\hat{Q}(s',a')\right].
\end{aligned}$$

$$\tag{15}$$

Then we introduce some intermediate terms and obtain

$$Q^*(s,a) - \hat{Q}(s,a)$$
$$= \gamma \sum_{s',a'} P(s'|s,a)\big[\pi^*(a'|s')Q^*(s',a') - \pi^*(a'|s')\hat{Q}(s',a') + \pi^*(a'|s')\hat{Q}(s',a') - \hat{\pi}(a'|s')\hat{Q}(s',a')\big]$$
$$= \gamma \sum_{s',a'} P(s'|s,a)\pi^*(a'|s')\big[Q^*(s',a') - \hat{Q}(s',a')\big] + \gamma \sum_{s',a'} P(s'|s,a)\hat{Q}(s',a')\big[\pi^*(a|s) - \hat{\pi}(a|s)\big]$$
$$\leq \gamma\|Q^* - \hat{Q}\|_\infty + \gamma\|P\hat{Q}\|_\infty\|\pi^* - \hat{\pi}\|_1 \qquad\qquad \triangleright \text{Hölder's inequality}$$
$$\leq \gamma\|Q^* - \hat{Q}\|_\infty + 2\gamma\|\hat{Q}\|_\infty \epsilon_\pi$$
$$\leq \gamma\|Q^* - \hat{Q}\|_\infty + \frac{2\gamma r_{\max}}{1-\gamma}\epsilon_\pi. \qquad\qquad \triangleright \hat{Q}(s,a) \leq \frac{r_{\max}}{1-\gamma}, \forall s,a$$

$$(16)$$

Thus,

$$\|Q^* - \hat{Q}\|_\infty - \gamma\|Q^* - \hat{Q}\|_\infty \leq \frac{2\gamma r_{\max}}{1-\gamma}\epsilon_\pi$$
$$(1-\gamma)\|Q^* - \hat{Q}\|_\infty \leq \frac{2\gamma r_{\max}}{1-\gamma}\epsilon_\pi.$$

$$(17)$$

Then we have

$$\|Q^* - \hat{Q}\|_\infty \leq \frac{2\gamma r_{\max}}{(1-\gamma)^2}\epsilon_\pi, \qquad\qquad\qquad (18)$$

which completes our proof. $\qquad\square$

Next, we consider the value discrepancy bound in policy imitation with the knowledge gap. Specifically, we denote the optimal value function with oracle states $s^o = [s^l, s^p]$ as $Q^*(s^l, s^p, a)$, and the value function with local states as $Q(s^l, a)$. Then we have the following discrepancy bound (restate of Theorems 1).

**Theorem A.2** (Policy imitation discrepancy). *The value discrepancy between the optimal value function with privileged knowledge and the value function with local state is bounded as*

$$\sup_{s^l, s^p, a} \big|Q^*(s^l, s^p, a) - \hat{Q}^{\hat{\pi}}(s^l, a)\big| \leq \frac{2\gamma r_{\max}}{(1-\gamma)^2}\epsilon_{\hat{\pi}}, \qquad (19)$$

*where*

$$\epsilon_{\hat{\pi}} = \sup_{s^l, s^p} D_{\mathrm{TV}}(\pi^*(\cdot|s^l, s^p)\|\hat{\pi}(\cdot|s^l)) \qquad\qquad (20)$$

*is the policy divergence between $\pi^*$ and $\hat{\pi}$, and $r_{\max}$ is the maximum reward in each step.*

*Proof.* We note that the transition functions for learning $\hat{\pi}(a|s^l)$ and $\pi^*(a|s^l, s^p)$ are the same and follow the ground-truth dynamics. Although we cannot observe the privileged state $s^p$ in learning the local policy, the transition of the next state $P(s^l_{t+1}, s^o_{t+1}|s^l_t, s^p_t)$ still depends on the oracle state of the previous step that contains the privileged state $s^p_t$. Since the agent is directly interacting with the environment, the privileged state does affect the transition functions. As a result, for policy imitation of the local policy, we have a similar derivation as in (15) because the transition function in (15) is the same for $\pi^*(a_{t+1}|s^o_{t+1})Q^*(s^o_{t+1}, a_{t+1})$ and $\hat{\pi}(a_{t+1}|s^l_{t+1})\hat{Q}(s^l_{t+1}, a_{t+1})$, which leads to a similar discrepancy bound as in an ordinary MDP. $\qquad\square$

### A.3   Proof of Theorem 2

We have defined the history encoder $\hat{P}_\phi$ which takes the history as input and the privileged state as output:

$$\hat{s}^p_t = \hat{P}(\cdot|h_t). \qquad\qquad\qquad (21)$$

In the following, we give the performance discrepancy bound between the optimal value function with oracle state $s^o_t = [s^l_t, s^p_t]$ and the value function with local state $s^l_t$ as well as the predicted privileged state $\hat{s}^p_t$ (restate of Theorem 2).

**Theorem A.3** (Privilege modeling discrepancy). *Let the divergence between the privileged state model $\hat{P}(s^p_{t+1}|h_{t+1})$ and the true distribution of privileged state $P(s^p_{t+1}|h_{t+1})$ be bounded as*

$$\epsilon_{\hat{P}} = \sup_{t \geq t_0} \sup_{h_{t+1}} D_{\text{TV}}\big(P(\cdot \mid h_{t+1}) \,\|\, \hat{P}(\cdot \mid h_{t+1})\big). \tag{22}$$

*Then the performance discrepancy bound between the optimal value function with $P$ and the value function with $\hat{P}$ holds, as*

$$\sup_{t \geq t_0} \sup_{s^l, s^p, a} |Q^*(s^l_t, s^p_t, a_t) - \hat{Q}_t(s^l_t, \hat{s}^p_t, a_t)| \leq \frac{\Delta_{\mathbb{E}}}{(1-\gamma)} + \frac{2\gamma r_{\max}}{(1-\gamma)^2}\epsilon_{\hat{P}}, \tag{23}$$

*where $\Delta_{\mathbb{E}} = \sup_{t \geq t_0} \big\|Q^* - \mathbb{E}_{s^p_t \sim P(\cdot|h_t)}[Q^*]\big\|_\infty + \big\|\hat{Q} - \mathbb{E}_{\hat{s}^p_t \sim \hat{P}(\cdot|h_t)}[\hat{Q}]\big\|_\infty$ is the difference in the same value function with sampled $s^p_t$ and the expectation of $s^p_t$ conditioned on $h_t$.*

*Proof.* For $\hat{Q}$ that estimates the value function of state $(s^l_t, \hat{s}^p_t)$, the value is stochastic since the hidden prediction $\hat{s}^p_t$ is a single sample from the output of model $\hat{P}(\cdot|h_t)$. We take the expectation to the output of $\hat{P}(\cdot|h_t)$ and obtain the expected $\hat{Q}$ as

$$\mathbb{E}_{\hat{s}^p_t \sim \hat{P}(\cdot|h_t)}[\hat{Q}_t(s^l_t, \hat{s}^p_t, a_t)]$$
$$= \mathbb{E}_{s^p_t \sim P(\cdot|h_t)}[r(s^l_t, s^p_t)] + \gamma \mathbb{E}_{s^l_{t+1} \sim P(\cdot|h_t,a_t), \hat{s}^p_{t+1} \sim \hat{P}(\cdot|h_t,a_t,s^l_{t+1})}\big[\max_{a'} \hat{Q}_{t+1}(s^l_{t+1}, \hat{s}^p_{t+1}, a')\big] \quad \triangleright (21)$$
$$= \mathbb{E}_{s^p_t \sim P(\cdot|h_t)}[r(s^l_t, s^p_t)] + \gamma \mathbb{E}_{s^l_{t+1} \sim P(\cdot|h_t,a_t), \hat{s}^p_{t+1} \sim \hat{P}(\cdot|h_{t+1})}\big[\max_{a'} \hat{Q}_{t+1}(s^l_{t+1}, \hat{s}^p_{t+1}, a')\big], \tag{24}$$

where $h_{t+1} = h_t \cup \{s^l_{t+1}, a_t\}$. We remark that the reward function $r(s^l_t, s^p_t)$ is returned by the real environment, thus $s^p_t$ follows the true $P$ in the reward.

**Remark 1** (Explanation of the Bellman Equation). *Our goal is to fit the Bellman equation in* (24). *Fitting the Bellman equation in* (24) *has several benefits. Firstly, it is easily implemented based on the fitted privileged state model $\hat{P}$. Specifically, solving* (24) *is almost the same as solving an ordinary MDP, with $\hat{s}^p_t$ generated by $\hat{P}$ based on the history in place of the true privileged state of $s^p_t$ (which is unavailable). Secondly and most importantly, when the history is sufficiently strong in predicting $s^p_t$, solving* (24) *leads to approximately optimal solutions, as we show in this theorem.*

For $Q^*$ that represents the optimal value function with the true privileged state, we have

$$Q^*(s^l_t, s^p_t, a_t) = r(s^l_t, s^p_t) + \gamma \mathbb{E}_{(s^l_{t+1}, s^p_{t+1}) \sim P(\cdot|s^l_t, s^p_t, a_t)}[\max_{a'} Q^*_{t+1}(s^l_{t+1}, s^p_{t+1}, a')]$$
$$= r(s^l_t, s^p_t) + \gamma \mathbb{E}_{(s^l_{t+1}, s^p_{t+1}) \sim P(\cdot|h_t, s^p_t, a_t)}[\max_{a'} Q^*_{t+1}(s^l_{t+1}, s^p_{t+1}, a')], \tag{25}$$

where the second equation holds since $h_t$ contains $s^l_t$. Then we take a similar expectation to the true hidden transition function $P$ as

$$\mathbb{E}_{s^p_t \sim P(\cdot|h_t)}[Q^*(s^l_t, s^p_t, a_t)]$$
$$= \mathbb{E}[r(s^l_t, s^p_t)] + \gamma \mathbb{E}_{(s^l_{t+1}, s^p_{t+1}) \sim P(\cdot|h_t, s^p_t, a_t), s^p_t \sim P(\cdot|h_t)}[\max_{a'} Q^*_{t+1}(s^l_{t+1}, s^p_{t+1}, a')]$$
$$= \mathbb{E}[r(s^l_t, s^p_t)] + \gamma \mathbb{E}_{(s^l_{t+1}, s^p_{t+1}) \sim P(\cdot|h_t, a_t)}[\max_{a'} Q^*_{t+1}(s^l_{t+1}, s^p_{t+1}, a')] \quad \triangleright \text{ according to Fig. 7}$$
$$= \mathbb{E}[r(s^l_t, s^p_t)] + \gamma \mathbb{E}_{s^l_{t+1} \sim P(\cdot|h_t, a_t), s^p_{t+1} \sim P(\cdot|h_t, a_t, s^l_{t+1})}[\max_{a'} Q^*_{t+1}(s^l_{t+1}, s^p_{t+1}, a')]$$
$$= \mathbb{E}[r(s^l_t, s^p_t)] + \gamma \mathbb{E}_{s^l_{t+1} \sim P(\cdot|h_t, a_t), s^p_{t+1} \sim P(\cdot|h_{t+1})}[\max_{a'} Q^*_{t+1}(s^l_{t+1}, s^p_{t+1}, a')], \tag{26}$$

where the last equation follows $h_{t+1} = h_t \cup \{s^l_{t+1}, a_t\}$, and the second equation follows the causal graph shown in Fig. 7. Specifically, following the relationship in Fig. 7, we have

$$P(s^o_{t+1}, h_t, s^p_t, a_t) = P(h_t)P(s^p_t|h_t)P(a_t|h_t)P(s^o_{t+1}|h_t, s^p_t, a_t). \tag{27}$$

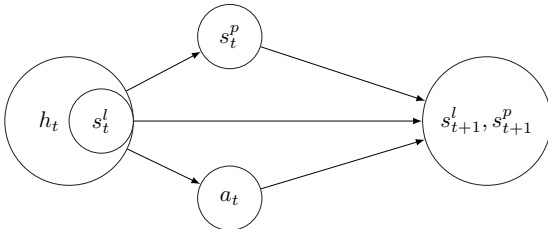

Figure 7: The causal graph. The agent takes action $a_t$ by only considering local states without accessing the privileged state $s_t^p$. The privileged state is based on history $h_t$. The next-state is conditional on $h_t$, $s_t^p$, and $a_t$.

Then we have

$$
\begin{aligned}
P(s_{t+1}^o|h_t, s_t^p, a_t)P(s_t^p|h_t) &= \frac{P(s_{t+1}^o, h_t, s_t^p, a_t)}{P(h_t)P(s_t^p|h_t)P(a_t|h_t)}P(s_t^p|h_t) = \frac{P(s_{t+1}^o, h_t, s_t^p, a_t)}{P(h_t)P(a_t|h_t)} \\
&= \frac{P(s_{t+1}^o, h_t, s_t^p, a_t)}{P(a_t, h_t)} = P(s_{t+1}^o, s_t^p|h_t, a_t),
\end{aligned}
\tag{28}
$$

where we denote $s_{t+1}^o = [s_{t+1}^l, s_{t+1}^p]$.

Based on above analysis, we derive the value discrepancy bound as follows. For $\forall s_t^l, s_t^p, a_t$, the discrepancy between $Q^*(s_t^l, s_t^p, a_t)$ and $\hat{Q}_t(s_t^l, \hat{s}_t^p, a_t)$ can be decomposed as

$$
\begin{aligned}
&Q^*(s_t^l, s_t^p, a_t) - \hat{Q}_t(s_t^l, \hat{s}_t^p, a_t) \\
&= \underbrace{Q^*(s_t^l, s_t^p, a_t) - \mathbb{E}_{s_t^p \sim P(\cdot|h_t)}[Q^*(s_t^l, s_t^p, a_t)]}_{(i)\Delta_{\mathbb{E}}^*} \underbrace{-\hat{Q}_t(s_t^l, \hat{s}_t^p, a_t) + \mathbb{E}_{\hat{s}_t^p \sim \hat{P}(\cdot|h_t)}[\hat{Q}_t(s_t^l, \hat{s}_t^p, a_t)]}_{(ii)-\hat{\Delta}_{\mathbb{E}}} \\
&+ \underbrace{\mathbb{E}_{s_t^p \sim P(\cdot|h_t)}[Q^*(s_t^l, s_t^p, a_t)] - \mathbb{E}_{s_t^p \sim P(\cdot|h_t)}[\hat{Q}_t(s_t^l, s_t^p, a_t)]}_{(iii)\ \text{value error}} \\
&+ \underbrace{\mathbb{E}_{s_t^p \sim P(\cdot|h_t)}[\hat{Q}_t(s_t^l, s_t^p, a_t)] - \mathbb{E}_{\hat{s}_t^p \sim \hat{P}(\cdot|h_t)}[\hat{Q}_t(s_t^l, \hat{s}_t^p, a_t)]}_{(iv)\ \text{model error}},
\end{aligned}
\tag{29}
$$

where we add three terms with positive sign and negative sign, including $Q^*$ with the expectation of $P(\cdot|h_t)$, $\hat{Q}_t$ with the expectation of $P(\cdot|h_t)$, and $\hat{Q}_t$ with the expectation of $\hat{P}(\cdot|h_t)$.

For $(i)$ and $(ii)$, we take the absolute value and get $\Delta_{\mathbb{E}}^* - \hat{\Delta}_{\mathbb{E}} \le |\Delta_{\mathbb{E}}^*| + |\hat{\Delta}_{\mathbb{E}}| = \Delta_{\mathbb{E}}$, where

$$
\begin{aligned}
\Delta_{\mathbb{E}}^* - \hat{\Delta}_{\mathbb{E}} &\le |\Delta_{\mathbb{E}}^*| + |\hat{\Delta}_{\mathbb{E}}| \le \sup_{s_t^l, s_t^p, a} \left(|\Delta_{\mathbb{E}}^*| + |\hat{\Delta}_{\mathbb{E}}|\right) \\
&= \sup_{s_t^l, s_t^p, a} \left|Q^*(s_t^l, s_t^p, a) - \mathbb{E}_{s_t^p \sim P(\cdot|h_t)}[Q^*(s_t^l, s_t^p, a)]\right| + \left|\hat{Q}_t(s_t^l, s_t^p, a) - \mathbb{E}_{\hat{s}_t^p \sim \hat{P}(\cdot|h_t)}[\hat{Q}_t(s_t^l, \hat{s}_t^p, a)]\right| \\
&= \left\|Q^* - \mathbb{E}_{s_t^p \sim P(\cdot|h_t)}[Q^*]\right\|_\infty + \left\|\hat{Q} - \mathbb{E}_{\hat{s}_t^p \sim \hat{P}(\cdot|h_t)}[\hat{Q}]\right\|_\infty,
\end{aligned}
\tag{30}
$$

which represents the difference in the same value function caused by the expectation of $s_t^p$ with respect to the history.

For $(iii)$ that represents the value difference in the optimal value function $Q^*$ and the current value function $\hat{Q}$ with the same distribution of state-action, we have the following bound as

$$\mathbb{E}_{s_t^p \sim P(\cdot|h_t)}[Q^*(s_t^l, s_t^p, a_t)] - \mathbb{E}_{s_t^p \sim P(\cdot|h_t)}[\hat{Q}_t(s_t^l, s_t^p, a_t)]$$

$$= \mathbb{E}[r(s_t^l, s_t^p)] + \gamma \mathbb{E}_{s_{t+1}^l \sim P(\cdot|h_t,a_t), s_{t+1}^p \sim P(\cdot|h_{t+1})}[\max_{a'} Q^*(s_{t+1}^l, s_{t+1}^p, a')]$$

$$\quad - \mathbb{E}[r(s_t^l, s_t^p)] + \gamma \mathbb{E}_{s_{t+1}^l \sim P(\cdot|h_t,a_t), s_{t+1}^p \sim P(\cdot|h_{t+1})}[\max_{a''} \hat{Q}_{t+1}(s_{t+1}^l, s_{t+1}^p, a'')] \qquad \triangleright \text{ from (26)}$$

$$= \gamma \mathbb{E}_{s_{t+1}^l \sim P(\cdot|h_t,a_t), s_{t+1}^p \sim P(\cdot|h_{t+1})}[\max_{a'} Q^*(s_{t+1}^l, s_{t+1}^p, a') - \max_{a''} \hat{Q}_{t+1}(s_{t+1}^l, s_{t+1}^p, a'')]$$

$$\leq \gamma \max_{a'} \mathbb{E}_{s_{t+1}^l \sim P(\cdot|h_t,a_t), s_{t+1}^p \sim P(\cdot|h_{t+1})}[Q^*(s_{t+1}^l, s_{t+1}^p, a') - \hat{Q}_{t+1}(s_{t+1}^l, s_{t+1}^p, a')]$$

$$\leq \gamma \sup_{s_{t+1}^l, s_{t+1}^p, a'} |Q^*(s_{t+1}^l, s_{t+1}^p, a') - \hat{Q}_{t+1}(s_{t+1}^l, s_{t+1}^p, a')|$$

$$= \gamma \|Q^* - \hat{Q}_{t+1}\|_\infty.$$

$$(31)$$

For $(iv)$ that represents the model difference of hidden state with different distributions, we have the following bound by following (24), as

$$\mathbb{E}_{s_t^p \sim P(\cdot|h_t)}[\hat{Q}_t(s_t^l, s_t^p, a_t)] - \mathbb{E}_{\hat{s}_t^p \sim \hat{P}(\cdot|h_t)}[\hat{Q}_t(s_t^l, \hat{s}_t^p, a_t)]$$

$$= \mathbb{E}[r(s_t^l, s_t^p)] + \gamma \mathbb{E}_{s_{t+1}^l \sim P(\cdot|h_t,a_t), s_{t+1}^p \sim P(\cdot|h_{t+1})}[\max_{a'} \hat{Q}_{t+1}(s_{t+1}^l, s_{t+1}^p, a')]$$

$$\quad - \mathbb{E}[r(s_t^l, s_t^p)] - \gamma \mathbb{E}_{s_{t+1}^l \sim P(\cdot|h_t,a_t), \hat{s}_{t+1}^p \sim \hat{P}(\cdot|h_{t+1})}[\max_{a''} \hat{Q}_{t+1}(s_{t+1}^l, \hat{s}_{t+1}^p, a'')]$$

$$= \gamma \mathbb{E}_{s_{t+1}^l \sim P(\cdot|h_t,a_t)} \left[ \mathbb{E}_{s_{t+1}^p \sim P(\cdot|h_{t+1})}[\max_{a'} \hat{Q}_{t+1}(s_{t+1}^l, s_{t+1}^p, a')] - \mathbb{E}_{\hat{s}_{t+1}^p \sim \hat{P}(\cdot|h_{t+1})}[\max_{a''} \hat{Q}_{t+1}(s_{t+1}^l, \hat{s}_{t+1}^p, a'')] \right]$$

$$= \gamma \mathbb{E}_{s_{t+1}^l \sim P(\cdot|h_t,a_t)} \sum_{s_{t+1}^p} \max_{a'} \hat{Q}_{t+1}(s_{t+1}^l, s_{t+1}^p, a') P(s_{t+1}^p|h_{t+1}) - \max_{a''} \hat{Q}_{t+1}(s_{t+1}^l, s_{t+1}^p, a'') \hat{P}(s_{t+1}^p|h_{t+1}).$$

$$(32)$$

Define a new function as $F_{t+1}(s_{t+1}^l, s_{t+1}^p) = \max_a \hat{Q}_{t+1}(s_{t+1}^l, s_{t+1}^p, a)$, then we have

$$\mathbb{E}_{s_t^p \sim P(\cdot|h_t)}[\hat{Q}_t(s_t^l, s_t^p, a_t)] - \mathbb{E}_{\hat{s}_t^p \sim \hat{P}(\cdot|h_t)}[\hat{Q}_t(s_t^l, \hat{s}_t^p, a_t)]$$

$$= \gamma \mathbb{E}_{s_{t+1}^l \sim P(\cdot|h_t,a_t)} \sum_{s_{t+1}^p} F_{t+1}(s_{t+1}^l, s_{t+1}^p) P(s_{t+1}^p|h_{t+1}) - F_{t+1}(s_{t+1}^l, s_{t+1}^p) \hat{P}(s_{t+1}^p|h_{t+1})$$

$$\leq \gamma \mathbb{E}_{s_{t+1}^l \sim P(\cdot|h_t,a_t)} \sum_{s_{t+1}^p} F_{t+1}(s_{t+1}^l, s_{t+1}^p) |P(s_{t+1}^p|h_{t+1}) - \hat{P}(s_{t+1}^p|h_{t+1})| \qquad (33)$$

$$\leq \gamma F_{\max} \|P(\cdot|h_{t+1}) - \hat{P}(\cdot|h_{t+1})\|_1 \qquad \triangleright \text{ Hölder's inequality}$$

$$= \frac{2\gamma r_{\max}}{1-\gamma} D_{\text{TV}}(P(\cdot|h_{t+1})\|\hat{P}(\cdot|h_{t+1})) \qquad \triangleright F_{\max} \leq r_{\max}/1-\gamma$$

According to derivation of $(i) \sim (iv)$, we take the supreme of $s_t^l, s_t^p, a_t$ and obtain the discrepancy between $Q^*(s_t^l, s_t^p, a_t)$ and $\hat{Q}_t(s_t^l, \hat{s}_t^p, a_t)$ as

$$\|Q^* - \hat{Q}_t\|_\infty \leq \sup_{s_t^l, s_t^p, a}(|\Delta_{\mathbb{E}}^*| + |\hat{\Delta}_{\mathbb{E}}|) + \frac{2\gamma r_{\max}}{1-\gamma} D_{\text{TV}}(P(\cdot|h_{t+1})\|\hat{P}(\cdot|h_{t+1})) + \gamma\|Q^* - \hat{Q}_{t+1}\|_\infty$$

$$\leq \sup_{s_t^l, s_t^p, a}(|\Delta_{\mathbb{E}}^*| + |\hat{\Delta}_{\mathbb{E}}|) + \frac{2\gamma r_{\max}}{1-\gamma} \epsilon_{\hat{P}_{t+1}} + \gamma\|Q^* - \hat{Q}_{t+1}\|_\infty$$

$$= \Delta_{\mathbb{E}}(t) + \frac{2\gamma r_{\max}}{1-\gamma} \epsilon_{\hat{P}_{t+1}} + \gamma\|Q^* - \hat{Q}_{t+1}\|_\infty,$$

$$(34)$$

where we define

$$\epsilon_{\hat{P}_{t+1}} = \sup_{h_{t+1}} D_{\text{TV}}(P(\cdot|h_{t+1})\|\hat{P}(\cdot|h_{t+1}))$$

as the model difference at time step $t+1$ with the worst-case history. Meanwhile, we define

$$\Delta_{\mathbb{E}}(t) = \sup_{s_t^l, s_t^p, a} \left( |\Delta_{\mathbb{E}}^*| + |\hat{\Delta}_{\mathbb{E}}| \right) = \underbrace{\left\| Q^* - \mathbb{E}_{s_t^p \sim P(\cdot|h_t)}[Q^*] \right\|_\infty}_{\text{(i)}} + \underbrace{\left\| \hat{Q} - \mathbb{E}_{\hat{s}_t^p \sim \hat{P}(\cdot|h_t)}[\hat{Q}] \right\|_\infty}_{\text{(ii)}},$$

(35)

to measure the difference in the same value function with true $s_t^p$ and the expectation of $s_t^p$ conditioned on history $h_t$.

In the following, we consider the step in worst case by taking supremum of $t$ when $t$ greater some $t_0$, and ensure the history is long and informative, as

$$\sup_{t \geq t_0} \|Q^* - \hat{Q}_t\|_\infty \leq \sup_{t \geq t_0} \Delta_{\mathbb{E}}(t) + \sup_{t \geq t_0} \frac{2\gamma r_{\max}}{1 - \gamma} \epsilon_{\hat{P}_{t+1}} + \gamma \sup_{t \geq t_0} \|Q^* - \hat{Q}_{t+1}\|_\infty$$

$$\leq \sup_{t \geq t_0} \Delta_{\mathbb{E}}(t) + \sup_{t \geq t_0} \frac{2\gamma r_{\max}}{1 - \gamma} \epsilon_{\hat{P}_{t+1}} + \gamma \sup_{t \geq t_0} \|Q^* - \hat{Q}_t\|_\infty$$

(36)

where we use inequality

$$\sup_{t \geq t_0} \|Q^* - \hat{Q}_{t+1}\|_\infty = \sup_{t \geq t_0+1} \|Q^* - \hat{Q}_t\|_\infty \leq \sup_{t \geq t_0} \|Q^* - \hat{Q}_t\|_\infty.$$

Then we have

$$\sup_{t \geq t_0} \|Q^* - \hat{Q}_t\|_\infty \leq \frac{\sup_{t \geq t_0} \Delta_{\mathbb{E}}(t)}{(1 - \gamma)} + \sup_{t \geq t_0} \frac{2\gamma r_{\max}}{(1 - \gamma)^2} \epsilon_{\hat{P}_{t+1}}$$

$$= \frac{\Delta_{\mathbb{E}}}{1 - \gamma} + \frac{2\gamma r_{\max}}{(1 - \gamma)^2} \epsilon_{\hat{P}},$$

(37)

where we define $\Delta_{\mathbb{E}} = \sup_{t \geq t_0} \Delta_{\mathbb{E}}(t)$ to consider the step of worst case by taking supremum of $t \geq t_0$. Meanwhile, we define the model error in worst case as

$$\epsilon_{\hat{P}} = \sup_{t \geq t_0} \epsilon_{\hat{P}}(t+1),$$

(38)

which can be minimized by using a neural network to represent $\hat{P}$, and reducing the error through Monte Carlo sampling of transitions in training.

We remark that in $\Delta_{\mathbb{E}}$ of Eq. (35), the term (i) in (35) captures how informative the history $h_t$ is in inferring necessary information of the hidden state $s_t^p$. Intuitively, term (i) characterizes the error in estimating the optimal $Q$-value with the informative history $h_t$. Similarly, term (ii) characterizes the error in predicting the hidden state $s_t^p$ that follows the estimated model $\hat{P}$ based on the history. If the model $\hat{P}$ is deterministic and each $\hat{s}_t^p \in \mathcal{S}^p$ has a corresponding history $h_t$, then term (ii) vanishes. Empirically, we use an informative history to make $\Delta_{\mathbb{E}}$ sufficiently small.

$\square$

# B  Environment Setting

We will open-source our code once the paper is accepted. Below, we provide the details of the environment setup.

## B.1  Details of Privileged DMC Benchmark

Based on existing benchmarks [57, 56], we develop three environments for privileged knowledge distillation, including *pendulum*, *finger spin*, and *quadruped walk*, that cover various training difficulties from easy to difficult. Each environment defines a series of privilege parameters, which are indeed real-world physical properties, such as gravity, friction, or mass of an object. Such privilege parameters define the behavior of the environments, and are only visible in training and hidden in testing.

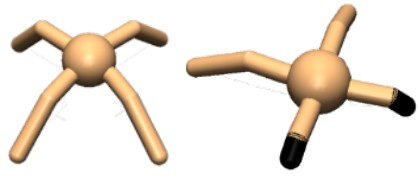

(a) origin        (b) morphology shift

Figure 8: An example of morphology shift.

In order to train a generalizable policy, we follow the domain randomization method and define a randomization range for privilege parameters for training. The underlying transition dynamics will change since we sample these privilege parameters uniformly from a pre-defined randomization range every episode. For policy evaluation, we additionally define two testing randomization ranges (*o.o.d* and *far o.o.d*) that contain out-of-distribution parameters compared to the training range. The privilege parameters and the corresponding ranges of each environment are listed in Table 4, Table 5, and Table 6, respectively.

In addition to the aforementioned shifted dynamic parameters, which belong to external privilege, we consider shifted morphology parameters, classified as internal privilege, in the *quadruped walk* task. Specifically, we reset the lengths of the front legs or back legs at the beginning of each episode. The lengths are sampled uniformly from the ranges indicated in Table 7. Visualization of morphology shift can be seen in Fig. 8.

Table 4: Privileged knowledge randomization range on *pendulum*. The testing range (ordinary) is the same as the training range. The privilege parameter $d_t$ refers to the observation interval length, $g$ to gravity, $l$ to the pole length, $m$ to the pole mass.

| Privilege Parameter | Training Range(ordinary) | Testing Range (o.o.d.) | Testing Range (far o.o.d.) |
|---|---|---|---|
| $d_t$ | [0.0, 0.05] | [0.0, 0.1] | [0.0, 0.2] |
| $g$ | [-2.0, 2.0] | [-3.0, 3.0] | [-4.0, 4.0] |
| $m$ | [-0.5, 0.5] | [-0.8, 0.8] | [-0.9, 1.5] |
| $l$ | [-0.5, 0.5] | [-0.8, 0.8] | [-0.9, 1.5] |

Table 5: Privileged knowledge randomization range on *finger spin*. The testing range (ordinary) is the same as the training range. friction_tangential, friction_torsional, and friction_rolling refer to the scaling factors for tangential friction, torsional friction, and rolling friction of all objects, respectively. $k_p$ and $k_d$ are the hyper-parameters for the PD controller for all joints. actuator_strength refers to the scaling factor for all actuators in the model. Viscosity refers to the scaling factor of viscosity used to simulate viscous forces. Wind_x, wind_y, and wind_z are used to compute viscous, lift and drag forces acting on the body, respectively.

| Privilege Parameter | Training Range(ordinary) | Testing Range (o.o.d.) | Testing Range (far o.o.d.) |
|---|---|---|---|
| friction_tangential | [-0.8, 0.8] | [-0.9, 1.0] | [-0.9, 1.0] |
| friction_torsional | [-0.8, 0.8] | [-0.9, 1.0] | [-0.9, 1.0] |
| friction_rolling | [-0.8, 0.8] | [-0.9, 1.0] | [-0.9, 1.0] |
| $k_p$ | [-0.8, 0.8] | [-0.9, 1.0] | [-0.9, 1.0] |
| $k_d$ | [0, 1.0] | [0.0, 1.5] | [0.0, 3.0] |
| actuator_strength | [-0.8, 0.8] | [-0.9, 1.0] | [-0.9, 1.0] |
| viscosity | [0, 0.8] | [0.0, 1.2] | [0.0, 2.0] |
| wind_x | [-2, 2] | [-3, 3] | [-7, 7] |
| wind_y | [-2, 2] | [-3, 3] | [-7, 7] |
| wind_z | [-2, 2 | [-3, 3]] | [-7, 7] |

Table 6: Privileged knowledge randomization range on *quadruped walk*. The testing range (ordinary) is the same as the training range. Density is used to simulate lift and drag forces, which scale quadratically with velocity. geom_density is used to infer masses and inertias. The other parameters have the same meaning described in Table 5.

| Privilege Parameter | Training Range(ordinary) | Testing Range (o.o.d.) | Testing Range (far o.o.d.) |
|---|---|---|---|
| gravity | [-0.2, 0.2] | [-0.4, 0.4] | [-0.6, 0.6] |
| friction_tangential | [-0.1, 0.1] | [-0.4, 0.4] | [-0.6, 0.6] |
| friction_torsional | [-0.1, 0.1] | [-0.4, 0.4] | [-0.6, 0.6] |
| friction_rolling | [-0.1, 0.1] | [-0.4, 0.4] | [-0.6, 0.6] |
| $k_p$ | [-0.1, 0.1] | [-0.4, 0.4] | [-0.6, 0.6] |
| $k_d$ | [0, 0.1] | [0.0, 0.4] | [0.0, 0.6] |
| actuator_strength | [-0.1, 0.1] | [-0.4, 0.4] | [-0.6, 0.6] |
| density | [0.0, 0.1] | [0.0, 0.4] | [0.0, 0.6] |
| viscosity | [0.0, 0.1] | [0.0, 0.4] | [0.0, 0.6] |
| geom_density | [-0.1, 0.1] | [-0.4, 0.4] | [-0.6, 0.6] |

Table 7: Privileged knowledge randomization range on *morph quadruped walk*. The testing range (ordinary) is the same as the training range. Length is used to identify the length of the modified leg. The other parameters have the same meaning described in Table 5.

| Privilege Parameter | Training Range(ordinary) | Testing Range (o.o.d.) | Testing Range (far o.o.d.) |
|---|---|---|---|
| gravity | [-0.2, 0.2] | [-0.4, 0.4] | [-0.6, 0.6] |
| friction_tangential | [-0.1, 0.1] | [-0.4, 0.4] | [-0.6, 0.6] |
| friction_torsional | [-0.1, 0.1] | [-0.4, 0.4] | [-0.6, 0.6] |
| friction_rolling | [-0.1, 0.1] | [-0.4, 0.4] | [-0.6, 0.6] |
| $k_p$ | [-0.1, 0.1] | [-0.4, 0.4] | [-0.6, 0.6] |
| $k_d$ | [0, 0.1] | [0.0, 0.4] | [0.0, 0.6] |
| actuator_strength | [-0.1, 0.1] | [-0.4, 0.4] | [-0.6, 0.6] |
| density | [0.0, 0.1] | [0.0, 0.4] | [0.0, 0.6] |
| viscosity | [0.0, 0.1] | [0.0, 0.4] | [0.0, 0.6] |
| geom_density | [-0.1, 0.1] | [-0.4, 0.4] | [-0.6, 0.6] |
| length | [-0.1, 0.1] | [-0.15, 0.15] | [-0.25, 0.25] |

## B.2 Details of Legged Robot Benchmark

We develop our codes based on Rudin et al. [3]. We create 4096 environment instances to collect data in parallel. In each environment, the robot is initialized with random poses and commanded to walk forward at $v_x^{cmd} = 0.4m/s$. The robot receives new observations and updates its actions every 0.02 seconds. Similar to Rudin et al. [3], we generate different sets of terrain, including smooth slope, rough slope, stairs up, stairs down, wave, and discrete obstacle (see Fig. 9 for visualization), with varying difficulty terrain levels. At training time, the environments are arranged in a $10 \times 20$ matrix with each row having terrain of the same type and difficulty increasing from left to right. We train

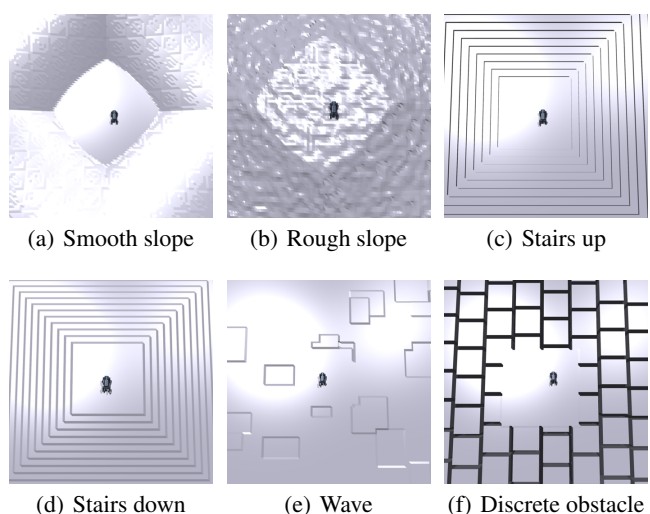

(a) Smooth slope     (b) Rough slope     (c) Stairs up

(d) Stairs down     (e) Wave     (f) Discrete obstacle

Figure 9: Visualization of stairs up, smooth slope, rough slope, stairs up, stairs down, and discrete obstacle.

with a curriculum over terrain where robots are first initialized on easy terrain and promoted to harder terrain if they traverse more than half its length. They are demoted to easier terrain if they fail to travel at least half the commanded distance $v_x^{cmd}T$ where $T$ is the maximum episode length.

The parameters of dynamic and heights of surroundings are defined as privilege parameters in this environment, as they can only be accessed in simulation but masked in the real world. The Heights of surroundings is an ego-centric map of the terrain around the robot. In particular, it consists of the height values $m_t = \{h(x, y)|(x, y) \in P\}$ at 187 points $P = \{-0.8, -0.7, \ldots, 0.7, 0.8\} \times \{-0.5, -0.4, \ldots, 0.4, 0.5\}$. The randomization range of dynamic parameters is shown in Table 8. We use the same reward function for our method and all baselines, including task rewards following [3] and style rewards (given by Adversarial Motion Priors [62]) following [63, 64].

Table 8: Privileged knowledge randomization range on Legged Robot Benchmark

| Privilege Parameter | Training Range | Testing Range |
|---|---|---|
| Added mass (kg) | [0, 3] | [0, 7] |
| $k_p$ | [22.4, 33.6] | [20, 60] |
| $k_d$ | [0.56, 0.84] | [0.5, 1.0] |

Table 9: Hyperparameters used in HIB

| HyperParameter | Value |
|---|---|
| $\tau$ | 0.01 |
| update_target_interval | 1 |
| $\lambda_1$ | 0.1 |
| $\lambda_2$ | 0.1 |
| history length $k$ | 50 |

## B.3 Details of Experiments on Real A1 Robot

We apply our method HIB, and the strongest baseline, DreamWaQ, which uses asymmetric actor critic without using HIB, to a Unitree A1 robot without any fine-tuning in the real world. The computations are performed on an onboard NVIDIA Jetson TX2. The policy runs at 50Hz and the target joint angles were tracked by a PD controller at a frequency of 200 Hz. The PD gains are $k_p = 28$ and $k_d = 0.7$, respectively. Policies rely on only the proprioception without any visual information for taking actions.

During the experiments, we send 2-dimensional linear velocity and 1-dimensional angular velocity commands to the robot with a remote controller. The performance of the robot is evaluated on different terrains, including plains, stairs, slopes and grass fields. The robot is commanded to perform basic moving skills like walking forward/backward and steering with different speed (maximum 1m/s)(see Fig. 10 for examples of real world terrains).

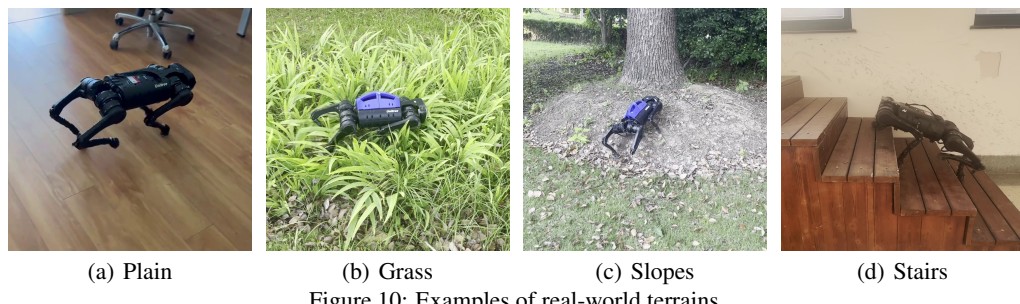

(a) Plain      (b) Grass      (c) Slopes      (d) Stairs

Figure 10: Examples of real-world terrains.

To further evaluate the robustness and the generalizability of the trained policy and also validate the effectiveness of the HIB method, we design two hard cases to test its performance. For the first case, the robot is required to safely jump down from a 0.6m-high stair. HIB can go down this stair with a $100\%$ success rate while DreamWaQ completely fails. For the second case, the robot is required to maintain its balance when a leg is suddenly pulled back in dashing. HIB can achieve $75\%$ success rate in 10 trials while DreamWaQ only obtains a $25\%$ success rate. See Fig. 11 for examples.

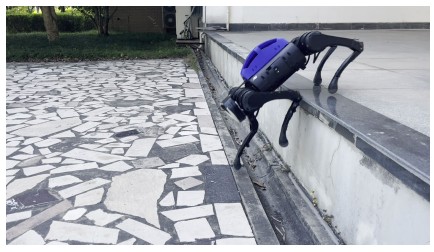
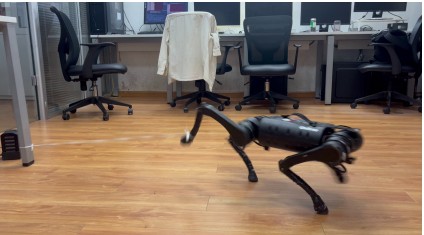

| (a) High stairs | (b) Pull a leg in dash |

Figure 11: Examples of two hard cases.

---

**Algorithm 1** Historcal Information Bottleneck (HIB)

---

*# Training Process (sim)*
**Initialize**: Buffer $\mathcal{D} = \{[s_t^l, s_t^p], a_t, r_t, [s_{t+1}^l, s_{t+1}^p], h_t\}$
**Initialize**: Historical encoder $f_\psi$, privilege encoder $f_\omega$, projector $g_\theta$ and target projector $g_{\theta^-}$.

1: **while** *not coverge* **do**
2:     Interact to the environment to collect $(s_i^o, a_i, r_i, s_{i+1}^o)$ with privileged state and save it to $\mathcal{D}$
3:     **for** $j$ from 0 **to** $N$ **do**
4:         Sample a batch of $(s_i^o, a_i, r_i, s_{i+1}^o)$ with history $h_i$
5:         Feed online $z_i \sim f_\psi(h_i)$, $y_i \leftarrow g_\theta(z_i)$, and target network $\tilde{e}_i \leftarrow f_\omega(s_i^p)$, $\tilde{y}_i \leftarrow g_{\theta^-}(\tilde{e}_i)$
6:         Compute cosine similarity $\mathcal{L}_{\text{sim}}(y_i, \tilde{y}_i)$, KL regularization $\mathcal{L}_{\text{KL}}$, reconstruction loss $\mathcal{L}_{\text{rec}}$ and RL objective $\mathcal{L}_{\text{RL}}$
7:         Update the HIB parameters via Eq. (13)
8:     **end for**
9: **end while**
*# Evaluation Process (real)*
1: Reset the environment and obtain the initial $s_0$ and $h_0$
2: **for** $t$ from $t_0$ **to** $t_{\max}$ **do**
3:     Estimate the privileged state via $z_t \sim f_\psi(h_t)$, and select action via $a \sim \hat{\pi}(a|s_t^l, z_t)$
4:     Interact with the environment, obtain the next state, set $s_t^l \leftarrow s_{t+1}^l$, and update $h_t$
5: **end for**

---

## C  Implementation Details

### C.1  Algorithm Implementation

Empirically we find that the learned privilege representation $z_t$ is requested to have similar and compact dimensions compared with another policy input $s_t^l$. So when privileged state $s_t^p$ is high-dimensional, for example, $s_t^p$ in Legged Robot benchmark, the encoder $f_\omega$ is necessary to project privileged state in a lower dimension, keeping the dimension between $z_t$ and $\tilde{e}_t$ being the same. In order to prevent information loss, we implement $f_\omega$ as an AutoEncoder [53]. Specifically, after $f_\omega$ projecting $s_t^p$ into representation $\tilde{e}_t$, we have a decoder denoted as $D_{\omega'}$ to reconstruct the $s_t^p$. Therefore, we leverage the following reconstruction loss to update encoder $f_\omega$:

$$\mathcal{L}_{\text{rec}} = \|s_t^p - D_{\omega'}(f_\omega(s_t^p))\|^2. \tag{39}$$

As a result, we employ Adam optimizer [55] to perform the following stochastic optimization step:

$$\omega \leftarrow \text{optimizer}(\omega, \nabla_\omega \mathcal{L}_{\text{rec}}). \tag{40}$$

In the Legged Robot benchmark, the privileged state consists of elevation information and some dynamic parameters, which are noisy and high-dimensional. To project $s_t^p$ in a lower-dimensional space, encoder $f_\omega$ can be a simple two-layer MLP. In the DMC benchmark, encoder $f_\omega$ can be an identity operator because the privileged state is defined as dynamic parameters that are compact and have similar dimensions with $s_t^l$. In this case, $z_t$ has the same dimensions as $s_t^p$ and $\tilde{e}_t$.

Considering there is no historical trajectory at the beginning of the episode, we warm up it with zero padding and then gradually update it for practical implementation. In order to compute HIB loss

more accurately, we maintain another IB buffer $\mathcal{B}_{\mathrm{ib}}$ to sample $h_t$ and $s_t^p$ practically, which only stores normal trajectories without zero padding.

## C.2 Model Architecture and Hyperparameters

- History encoder $f_\psi$ is a TCN model, consisting of three conv1d layers and two linear layers.
- The projector and target projector both have the same architecture, which is a two-layered MLP.
- Encoder $f_\omega$ is a two-layer MLP on the Legged Robot Benchmark, and an identical operator on the privileged DMC benchmark.
- Actor-Critic is 3-layered MLPs (with Relu activation).
- Hyperparameters used in HIB can be found in Table 9.

