# OpenReview forum: "Bridging the Sim-to-Real Gap from the Information Bottleneck Perspective"
_robot-learning.org/CoRL/2024/Conference — CoRL 2024_

### Official Review · Reviewer_jvAp · 2024-07-21
**novel and solid work**

**Originality:** 4
**Technical Quality:** 3
**Clarity Of Presentation:** 4
**Potential Impact:** 4
**Recommendation:** 3
**Confidence:** 4

**Review:**

To address the limitation in existing policy distillation framework which is computational expensive, lack of theoretical guarantee, and do not fully use privileged information. This paper introduces the Historical Information Bottleneck (HIB) method for reducing the sim-to-real gap in reinforcement learning. By leveraging historical data, the method optimizes the representation of privileged information, aiming to enhance real-world decision-making. In my view, this paper seems to be novel and technically solid. However, I have no time to go through the proofs of theorem 1&2.

While the paper presents a promising approach through the innovative use of the information bottleneck principle and historical data, there are several doubts about the technical parts:
1/ Around line 161-162, TV distance is convert to KL distance due to Pinsker's inequality. Reference is need.
2/ In equation 9, I cannot see how the second equality is derived.
3/ In section 5.2, around line 180, another notation \psi is introduce to distinguished from \phi. however, this f(.) seems do the same job map h_t to z_t. It is confusing to me.
4/ I find Z_t and z_t, S_t and s_t sometimes are used interchangeably. It would be better make it consistent.

The proposed method is tested in both simulation and real world. In my view, the real-world test should be more quantitive and with more well-design scenarios. Overall the paper is well written. The whole paper is a bit mathematically dense. The punctuation in equation (12) should be period.

**Quality Of The Limitations Section:**

3

**Questions For Rebuttal:**

see in the review.

**Robotics Focus:**

3

**Summary Of Paper:**

To address the limitation in existing policy distillation framework which is computational expensive, lack of theoretical guarantee, and do not fully use privileged information. This paper introduces the Historical Information Bottleneck (HIB) method for reducing the sim-to-real gap in reinforcement learning.

**Summary Of Recommendation:**

see in the review

---

### Official Review · Reviewer_HU57 · 2024-07-22

**Originality:** 3
**Technical Quality:** 3
**Clarity Of Presentation:** 3
**Potential Impact:** 2
**Recommendation:** 3
**Confidence:** 3

**Review:**

This paper introduces Historical Information Bottleneck (HIB), a new approach to make better use of privileged knowledge from simulations. It focuses on creating a compact representation of this knowledge to aid real-world decision-making by leveraging historical data that maximizes its connection to the privileged knowledge, reducing errors and improving performance.

Advantages:

1) Theoretical Analysis: It shows that minimizing privileged knowledge modeling errors leads to more effective policies.
2) Performance: HIB outperforms baseline methods, offering better generalization and robustness in both simulated and real-world tasks.

Issues:

1) For Figure 4, what happens if all baselines are trained for a longer period? Also, why does the vanilla RL method show such high variance? More seeds should be run to confirm whether the conclusions remain valid.
2) For Figure 1, the main training pipeline is unclear to readers who may not read the entire methods section. Consider revising the flow and style for better clarity, especially regarding gradient backpropagation and the different stages during training and deployment.

**Quality Of The Limitations Section:**

3

**Questions For Rebuttal:**

Some questions to address are:


1) Figure 4's Reproducibility: What are the results if all baselines are trained for a longer duration?  Running additional experiments with more seeds could help verify whether the observed conclusions are consistent and reliable.

2) Real-World Reward Function Definition: How is the reward function defined for real-world experiments?

3) Real-World Performance of RMA Baseline: What is the real-world performance of the RMA baseline? It's better to show how good/bad are others in the real world instead of showing HIB only.

**Robotics Focus:**

4

**Summary Of Paper:**

This paper introduces the Historical Information Bottleneck (HIB), a method that treats the sim-to-real transfer as an information bottleneck problem. HIB learns a privileged knowledge representation from historical data, improving generalizability and reducing value discrepancies between oracle and learned policies. Empirical results show that HIB outperforms previous methods in simulation tasks. The paper also shows HIB with good performance in real-world tasks.

**Summary Of Recommendation:**

In summary, I am leaning towards accepting the paper though it has some minor issues. Because it provides a detailed analysis of a knowledge distillation method for RL. The two-stream design and information bottleneck loss effectively reduce performance discrepancies, as demonstrated in both simulation and real-world experiments.

---

### Official Review · Reviewer_jExp · 2024-07-25
**Review for Bridging the Sim-to-Real Gap from the Information Bottleneck Perspective**

**Originality:** 3
**Technical Quality:** 4
**Clarity Of Presentation:** 3
**Potential Impact:** 3
**Recommendation:** 3
**Confidence:** 3

**Review:**

**Strengths**
1. The proposed method is theoretically sound approach to tackling the sim-to-real gap in RL. The paper provides a rigorous theoretical analysis, demonstrating how the proposed HIB method reduces the value discrepancy between oracle and learned policies.
2. The extensive experiments on simulated and real-world tasks show that HIB outperforms state-of-the-art methods. The real-world deployment on a quadrupedal robot has impressive performance, achieving a 100% success rate in navigating challenging terrains.

**Weaknesses**
1. As mentioned in the limitation section, this paper deals with limited scope of privileged knowledge: The method's effectiveness is constrained when dealing with high-dimensional and complex space such as RGB images. This limitation could restrict the method's applicability to a broader range of real-world scenarios.
2. Currently the paper deals with stationary historical data. In highly dynamic environments where historical data patterns change frequently, this assumption might not hold, potentially degrading performance.

**Quality Of The Limitations Section:**

3

**Questions For Rebuttal:**

1. Handling Multi-Modal Privileged Knowledge: As the authors have already pointed out in the limitation section, the method cannot handle multi-modal privileged knowledge such as RGB images. Are there any preliminary results or insights into this extension?
2. Impact of Historical Trajectory Length: How does the length of the historical trajectory impact the performance of the HIB method? Currently the method uses a fixed length. Is there an optimal length, and how sensitive is the method to variations in this parameter? More analysis on this will be appreciated
3. Noise and Uncertainty Handling: How does the HIB method handle noise and uncertainty in the historical data? Are there mechanisms to mitigate the impact of noisy or incomplete historical trajectories on the performance of the learned policy?

**Robotics Focus:**

4

**Summary Of Paper:**

The paper proposes a novel method called Historical Information Bottleneck (HIB) to address the sim-to-real gap in reinforcement learning. It assumes during training time, privileged knowledge is available in simulation, and agents only have access to local states in the real world. The proposed HIB method uses the Information Bottleneck principle to distill privileged knowledge from historical information by maximizing the MI between the historical representation and the privileged state via contrastive learning. The paper presents theoretical analysis and empirical results demonstrating HIB's efficacy in both simulated and real-world environments.

**Summary Of Recommendation:**

I recommend weak accept of the paper based on my evaluations.

---

### Official Review · Reviewer_iHnC · 2024-08-01
**A convincing approach to policy generalization for bridging the sim2real gap**

**Originality:** 3
**Technical Quality:** 4
**Clarity Of Presentation:** 3
**Potential Impact:** 3
**Recommendation:** 4
**Confidence:** 3

**Review:**

### Strengths

The approach is technically sound and the authors clearly explain the theoretical background behind underlying historical information bottleneck and its application to policy generalization.

The experimental results are strong, particularly the real-robot experiments clearly indicate robust behaviors in the real world.
The ablation studies and investigation of the learned privileged state representation is convincing that the proposed approach achieves more robust policy generalization than previous works.
The supplemental video is well produced and demonstrates the experimental setups as well as detailed real-robot evaluations.

### Weaknesses

The motivation behind the two-stream architecture should be clarified (see more points in my first question for rebuttal), especially implementation details, such as why stop gradient operations are needed. The architecture appears close to the one in [54] but it would be helpful to have a more self-contained presentation of the entire approach and design choices.

The limitations should be explained more clearly. It seems that the generalization success of this work hinges on the closeness of the local state spaces in simulation and reality, and the realism of the simulated dynamics of these states (simulation states need to closely follow the same dynamics of the real world states), i.e. the sim2real gap already has to be sufficiently small for the policy to generalize to the real world.

It would be better if the authors clarified in the main paper in Sec. 6.4 that the only evaluated baseline (even though it could be the best performing baseline) on the real robot was DreamWaQ instead of claiming "the baselines completely fail".

### Small issues:
* Typo in line 90: "fraction" instead of "friction"
* Line 99: the term "oracle" policy is perhaps not appropriate since it does not have access to oracle states
* Variable $r_\max$ in Eq. (2) should be explained
* Line 316: I assume the authors mean limited "in" instead of limited "to" in this context?

**Quality Of The Limitations Section:**

2

**Questions For Rebuttal:**

Why is it necessary to project $\tilde{e}_t$ and use this projection to compute cosine similarity with the projection of $z_t$? Why can the autoencoder of the target network not directly learn a latent variable $\tilde{z}_t$ whose cosine similarity to $z_t$ is maximized, i.e. without the projection step (given that both $\tilde{e}_t$  and $z_t$ already have the same dimension)?

Since the authors only mention the lack of support for recovering multi-modal privileged information, I wonder if there is a more significant limitation that should at least be mentioned:
How valid is the assumption that the local state space from the simulated environment is close enough for the policy to perform well in the real world?  Is it possible to quantify the closeness between such state representations (sim and real) within the proposed method is robust to generalize to the real world?

**Robotics Focus:**

4

**Summary Of Paper:**

The paper introduces Historical Information Bottleneck (HIB), a policy generalization approach that considers bridging the sim2real gap as an information bottleneck problem.   The method proposes to learn an encoding $f_\varphi$ of the privileged state (only knowledge which is accessible in simulation) conditioned on the history of local states $h_t$. Such history encoding yields a latent variable $z_t$ which the generalist policy is conditioned on, next to the current local state $s_t^l$. To maximize mutual information $I(Z_t; S_t^p)$ between the latent representation $Z_t$ and the privileged knowledge $S_t^p$, a contrastive objective is proposed. Simplifying such objective by removing negative sampling, the proposed method uses cosine similarity with only positive samples $z_t\sim f_\varphi(h_t)$, resulting in a two-stream architecture consisting of an online and a target network. The online network learns the latent encoding of privileged information from the history of local states via a time convolutional neural network (TCN), the target network learns a representation $\tilde{e}_t$ via an AutoEncoder of the privileged state. The cosine similarity between projections of both network outputs are used to evaluate $I(Z_t; S_t^p)$. At the same time, a regulizing loss term is used to ensure the information irrelevant to the privileged state information is compressed.  The authors demonstrate the performance of the proposed sim2real method on the DeepMind Control Suite benchmark and a real quadruped robot. Ablation studies over different HIB variants on the quadriped walking task are given. The learned latent representations are visualized through t-SNE which indicate that HIB is able to better recover the privileged information than previous policy distillation methods.

**Summary Of Recommendation:**

I tend to recommend the paper for acceptance. The approach is sound and the real robot results are strong. The paper is not a strong accept for me due to the remaining clarity issues and the potential limitation I raised in my review.

---

### Author Rebuttal · Authors · 2024-08-07

# General Response
We thank all of the reviewers for their time and insightful feedback! Furthermore, we are very glad to find that reviewers generally recognized our solid theoretical analysis and the superior performance of our method:
## Contributions:
- **Method**: The approach is technically sound [**iHnC**].The proposed method is theoretically sound approach to tackling the sim-to-real gap in RL [**jExp**]. The paper provides a rigorous theoretical analysis [**jExp**]. The paper presents a promising approach through the innovative use of the information bottleneck principle and historical data [**jvAp**].
- **Presentation**: The authors clearly explain the theoretical background [**iHnC**]. Overall the paper is well written [**jvAp**]. The supplemental video is well produced and demonstrates the experimental setups as well as detailed real-robot evaluations [**iHnC**].
- **Experiments**: The experimental results are strong [**iHnC**]. The extensive experiments on simulated and real-world tasks show that HIB outperforms state-of-the-art methods [**jExp,HU57**]. The real-world deployment on a quadrupedal robot has impressive performance [**jExp**]. HIB offers better generalization and robustness in both simulated and real-world tasks [**HU57**]. The ablation studies and investigation of the learned privileged state representation is convincing [**iHnC**].

We will carefully update our paper in the next version to incorporate the valuable suggestions of the reviewers. In summary, we add the following new experiments to address the reviewers' concerns:
- [**iHnC**] We visualize the closeness of the local state spaces in simulation and reality [see Figure 2 of the `additional_results.pdf`].
- [**jExp**] We ablate the length of the historical trajectory [ see figure 1 of `additional_results.pdf`]
- [**HU57**] We evaluate the performance of vanilla RL with more seeds [see figure 3 of `additional_results.pdf`].
- [**HU57**] We report the performance of the RMA baseline in the real-world hard cases [see the `additional_video.mp4` in the zip].
- [**jvAp**] We test the performance of HIB in more well-design scenarios [see the `additional_video.mp4` in the zip]

**Our zip file contains `additional_video.mp4` and `additional_results.pdf` for your review**. We hope to have addressed all the raised concerns and would be happy to respond to further questions and suggestions. We plan to open-source the codes upon acceptance for reproducibility and future research.

---

### Decision · Program_Chairs · 2024-09-04

**Decision:**

Accept

**Comment:**

The paper addresses the sim-to-real gap by proposing a privileged knowledge distillation method called the Historical Information Bottleneck (HIB) that captures the changeable dynamic information.

Strengths:
- Sound approach with a solid theoretical background
- Empirical results and comparisons are strong, particularly the real-robot experiments

Weaknesses:
- Lack of implementation details
- Limitations should be explained more clearly
- Limitations when dealing with high-dimensional observations
- Missing technical details of the method

The reviewers provided valuable suggestions to improve the paper. I encourage the authors to work with the reviewers in addressing their concerns.
=============================
Pos-rebuttal update

Thank you addressing the reviewers' comments. Great work!